# Deep Bayesian Active Learning for Preference Modeling in Large Language Models

Luckeciano C. Melo[*1,2]    Panagiotis Tigas[1]    Alessandro Abate[†2]    Yarin Gal[†1]
[1] OATML, University of Oxford    [2] OXCAV, University of Oxford

## Abstract

Leveraging human preferences for steering the behavior of Large Language Models (LLMs) has demonstrated notable success in recent years. Nonetheless, data selection and labeling are still a bottleneck for these systems, particularly at large scale. Hence, selecting the most informative points for acquiring human feedback may considerably reduce the cost of preference labeling and unleash the further development of LLMs. Bayesian Active Learning provides a principled framework for addressing this challenge and has demonstrated remarkable success in diverse settings. However, previous attempts to employ it for Preference Modeling did not meet such expectations. In this work, we identify that naive epistemic uncertainty estimation leads to the acquisition of redundant samples. We address this by proposing the **B**ayesian **A**ctive **L**earner for **P**reference **M**odeling (BAL-PM), a novel stochastic acquisition policy that not only targets points of high epistemic uncertainty according to the preference model but also seeks to maximize the entropy of the acquired prompt distribution in the feature space spanned by the employed LLM. Notably, our experiments demonstrate that BAL-PM requires 33% to 68% fewer preference labels in two popular human preference datasets and exceeds previous stochastic Bayesian acquisition policies.

## 1 Introduction

Preference Modeling is a key component to aligning unsupervised pre-trained Large Language Models (LLMs) towards human preferences [1–4]. It is often performed by collecting human feedback for a set of prompt-completion pairs and then leveraging the data to steer the behavior of such models, either directly [5] or via reward models [6]. Nevertheless, human feedback generation is laborious [7], especially when it requires specialized knowledge [8, 9]. Furthermore, the quality of the prompts has a crucial impact on the performance of fine-tuned models [10]. Hence, selecting the most informative points to gather feedback is essential to reduce costs and enable better LLMs.

Despite its substantial impact, data selection for Preference Modeling poses a significant challenge. The prompt-completion pool is arbitrarily

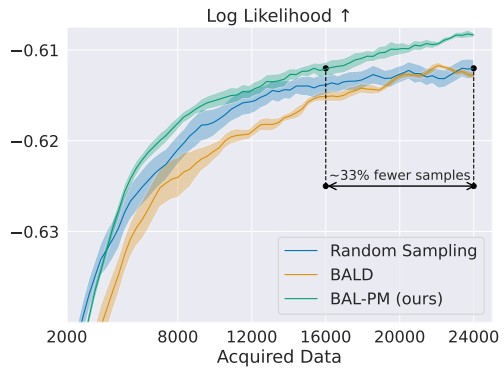

Figure 1: **Log-Likelihood of learned preference models in the Reddit TL;DR dataset** [1]. Our method, BAL-PM, reduces the volume of required human feedback by 33% over random acquisition.

---

[*]Correspondence to: luckeciano.carvalho.melo@cs.ox.ac.uk
[†]Denotes equal supervision.

38th Conference on Neural Information Processing Systems (NeurIPS 2024).

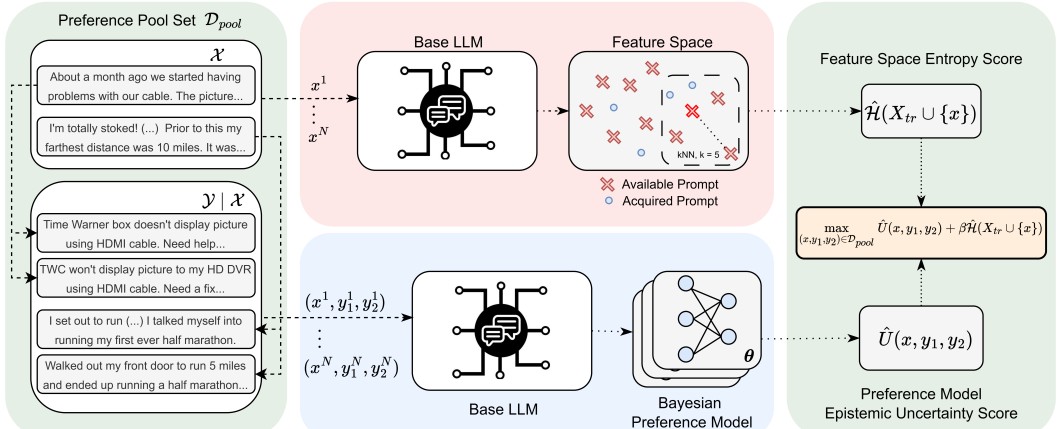

Figure 2: **An illustration of how BAL-PM works**. For each tuple $(x, y_1, y_2) \in \mathcal{D}_{pool}$, we obtain features for the prompt and prompt-completion pairs by computing the last layer embeddings of the base LLM. We leverage the prompt feature space to estimate the entropy score of the acquired prompt distribution, $\hat{\mathcal{H}}(X_{train} \cup \{x\})$. Similarly, we use the prompt-completion features as input for the Bayesian Preference Model, which is used to estimate task-dependent epistemic uncertainty scores, $\hat{U}(x, y_1, y_2)$. BAL-PM selects the tuple that maximizes the linear combination of both scores.

large and semantically rich. Additionally, human feedback is inherently noisy, with low agreement rates among labelers, typically observed between $60\% - 75\%$ for these settings [6, 1, 11, 12]. Lastly, the intrinsic scale of LLM development requires parallelized labeling and makes frequent model updates prohibitively expensive, limiting the applicability of many active learning schemes that rely on single-point acquisition [13].

Bayesian Active Learning provides a principled approach to data selection [14–16], which has demonstrated remarkable success across different fields [17–19]. However, its application in Active Preference Modeling is not straightforward. Past attempts of employing the framework in this setting reported no benefits over random selection [20], arguably due to poor uncertainty estimation in the context of LLMs, which is indeed an open challenge and active area of research [21].

We identify two reasons for this phenomenon. First, the inherent bias of approximate Bayesian inference in deep learning models, particularly for LLMs. Second, and more nuanced, the current intractability of epistemic uncertainty estimation methods in Preference Modeling for LLMs, a context that intrinsically requires batch acquisition. Proper estimators for this setting present combinatorial complexity, and even greedy approximations are still computationally demanding and impractical [13, 22]. This limitation leads to relying on simpler single-point acquisition schemes such as BALD [23] (as in Gleave and Irving [20]), designed to acquire individual points followed by model updates. However, these assumptions are far from realistic for the scale of Preference Modeling in LLMs, and naively applying such methods for batch acquisition leads to the selection of redundant samples.

In this work, we argue that leveraging the information available from the feature space spanned by the LLM – a task-agnostic[3] source of epistemic uncertainty – alleviates these problems. We propose **B**ayesian **A**ctive **L**earner for **P**reference **M**odeling (BAL-PM), a novel stochastic acquisition policy that not only targets points of high epistemic uncertainty according to the preference model but also seeks to maximize the entropy of the acquired prompt distribution in the feature space. This entropy score encourages the active learner to select prompts from low-density regions, effectively reducing the feature space epistemic uncertainty [24]. As a result, it promotes diversity in the acquired training set, preventing the selection of redundant samples and also helping in learning a better Bayesian preference model and its task-dependent epistemic uncertainty estimates for subsequent acquisitions. Figure 2 illustrates how BAL-PM works.

We conduct active learning experiments in the Reddit and CNN/DM preference datasets [25, 26, 1] to validate our method. BAL-PM demonstrates strong gains over random sampling, reducing by approximately $33\%$ (as shown in Figure 1) and $68\%$ the volume of feedback required to learn the

---

[3]"task" refers to Preference Modeling. A task-agnostic estimation is independent of preference labels.

preference model in the considered datasets. It also consistently surpasses other strong stochastic Bayesian acquisition policies [22]. Finally, we further analyze the acquired prompt distribution to show that BAE-PM prevents redundant exploration and effectively balances the contribution of the two sources of epistemic uncertainty.

## 2 Related Work

**Bayesian Active Learning** is an established form of active learning that leverages the uncertainty in model parameters to select the most informative points [14, 27, 16], demonstrating relevant impact in several applications [19, 18, 17, 28, 29]. In this work, we apply this technique for Preference Modeling [30, 31] in LLMs. Given the requirements of such a problem setting, we focus on batch acquisition [14, 13], particularly in the design of stochastic acquisition policies, similarly to Kirsch et al. [22]. However, our work fundamentally differs from theirs in the strategy of incorporating stochasticity. The policies introduced by Kirsch et al. [22] directly sample from the distribution determined by the single-point, task-dependent epistemic uncertainty scores. In contrast, our method maximizes the entropy of the acquired data distribution, which allows leveraging a task-agnostic source of epistemic uncertainty, alleviating the effect of biased task-dependent uncertainty scores.

**Active Preference Modeling** leverages active learning techniques to reduce the feedback needed for Preference Modeling [27]. There has been a recent surge of interest in the area [32–35, 20, 36] given the impact of Preference Optimization in fine-tuning LLMs [2, 5, 10, 6]. A portion of this line of work focuses on query generation to directly optimize preferences. Mehta et al. [32] theoretically formalizes the problem and proposes a method that generates one completion to maximize the uncertainty of the triple in a kernelized setting. Das et al. [35] proposes a method based on confidence bands, accounting for both completions in the triple and relaxing linearity assumptions on the reward function. Ji et al. [33] constructs an optimistic estimator for the reward gap between completions and selects those with the least gap, using an uncertainty estimator to reduce query complexity. Lastly, Dwaracherla et al. [36] generate completions using double Thompson Sampling, representing epistemic uncertainty with an Epistemic Neural Network [37], similar to our Bayesian model. Overall, while having the shared goal of reducing the volume of human feedback, query generation is orthogonal to our problem setting. Instead, we focus on the pool-based setting, as in Gleave and Irving [20], which allows us to leverage real human feedback in experiments rather than relying on synthetic preference simulators. Gleave and Irving [20] was the first attempt of Bayesian Active Learning in this setting, and it directly applies BALD acquisition [23] for Preference Modeling, using a fully fine-tuned deep ensemble for epistemic uncertainty estimation. In contrast, our work proposes a new objective that extends BALD acquisition to account for the entropy of the acquired prompt distribution to encourage the acquisition of more diversified samples, and formulates the Bayesian model as an ensemble of adapters.

**Task-Agnostic Uncertainty Estimation** refers to a set of techniques that quantifies uncertainties based on density estimation of the input in a learned latent feature space [38, 39]. In this context, distant points from the training set offer more information about the input space, which is useful for out-of-distribution detection [39] and unsupervised active learning [40]. Similarly, leveraging information from the feature space via entropy maximization is a common approach in Reinforcement Learning for state exploration [41–44]. While our method relies on the same principles – acquiring more information about the feature space – our problem setting and methodology differ substantially, as we focus on Active Preference Modeling in the context of LLMs.

## 3 Preliminaries

**Problem Statement**. We formulate our setup as an *active inverse* variant of the contextual dueling bandit problem [45, 46]. We assume a prompt space $\mathcal{X}$, an action space $\mathcal{Y}$, and a language policy $\tau : \mathcal{X} \times \mathcal{Y} \to [0, \infty)$. Given $x \sim \mathcal{X}$, this language policy $\tau$ (e.g., an LLM) selects actions $y_1, y_2 \sim \tau(\cdot \mid x)$ (also referred to as completions), generating a dataset of tuples $\mathcal{D}_{pool} = \{x^i, y_1^i, y_2^i\}^N$. Crucially, $x$ is sampled with replacement, i.e., we may generate multiple completions for the same prompt. Then, we define a policy $\pi : \mathcal{X} \times \mathcal{Y} \times \mathcal{Y} \to [0, \infty)$, which select tuples $(x, y_1, y_2) \in \mathcal{D}_{pool}$ to query for human binary preference over completions $y_1 \succ y_2$, forming a preference dataset $\mathcal{D}_{train} = \{x^i, y_1^i, y_2^i, y_1 \succ y_2^i\}^B$. Finally, $D_{train}$ is used to learn a preference model $p_{\boldsymbol{\theta}}(y_1 \succ y_2 \mid x, y_1, y_2)$, parameterized by $\boldsymbol{\theta}$, which aims to recover the human preference function. The goal is to find $\pi$ that minimizes the amount of samples $B$ required to learn $p_{\boldsymbol{\theta}}(y_1 \succ y_2 \mid x, y_1, y_2)$.

**Preference Modeling**. In this work, we assume that the preferences $y_1 \succ y_2$ are generated by an unknown latent reward model $r(x, y)$. We model $y_1 \succ y_2$ following the Bradley-Terry (BT) model [47]:

$$p(y_1 \succ y_2 \mid x, y_1, y_2) = \frac{\exp r(x, y_1)}{\exp r(x, y_1) + \exp r(x, y_2)}. \tag{1}$$

The BT model is often implemented by learning a parameterized latent reward model $r_{\boldsymbol{\theta}(x,y)}$ and optimizing $\boldsymbol{\theta}$ via maximum likelihood estimation. This means minimizing the negative Log-Likelihood with respect to the human preference labels.

**Bayesian Active Learning**. We adopt a Bayesian Model, which assumes a probability distribution over the parameters $\boldsymbol{\theta}$, such that, given a classification setting with inputs $x \sim X$ and labels $y \sim Y$ the predictive preference distribution is given by:

$$p(y \mid x) = \mathbb{E}_{p(\boldsymbol{\theta})}[p(y \mid x, \boldsymbol{\theta})]. \tag{2}$$

For active learning, we follow the methodology introduced by Lindley [23], namely BALD (Bayesian Active Learning by Disagreement), which proposes that the utility of a data point $x \sim X$ is given by the expected information gain about the parameters $\boldsymbol{\theta}$ with respect to the predictive distribution, a proxy of epistemic uncertainty:

$$U(x) \coloneqq \mathcal{I}(\boldsymbol{\theta}, y \mid \mathcal{D}_{train}, x) \quad = \mathcal{H}(p(y \mid \boldsymbol{x}, \mathcal{D}_{train})) - \mathbb{E}_{p(\boldsymbol{\theta} \mid \mathcal{D}_{train})}[\mathcal{H}(p(y \mid \boldsymbol{x}, \boldsymbol{\theta})]. \tag{3}$$

**Kozachenko–Leonenko Entropy**. The KL entropy estimator [48] is a non-parametric, particle-based estimator that leverages the $k$-nearest neighbors distance. Given a random variable $X$ and a set of $N$ i.i.d particles $\{x_i\}^N$, $x_i \sim X$, the KL entropy estimation for $X$ is defined as:

$$\hat{\mathcal{H}}_{KL}(X) = \frac{d_X}{N} \sum_{i=0}^{N} \log D_x(i) + \log v_{d_X} + \psi(N) - \psi(k), \tag{4}$$

where $d_X$ is the dimension of $X$, $v_{d_X}$ is the volume of the $d_X$-dimensional unit ball, $\psi$ is the digamma function, and $D_x(i)$ is twice the distance between the particle $x_i$ to its $k$-nearest neighbor.

## 4 Bayesian Active Learner for Preference Modeling

We now introduce our method for Active Preference Modeling, BAL-PM, illustrated in Figure 2. Our desiderata is to design an acquisition policy that addresses the shortcomings of naive epistemic uncertainty estimation – such as the acquisition of redundant samples – by leveraging an unsupervised source of epistemic uncertainty that encourages diversity in the acquired training distribution.

**Objective**. Based on the above, we propose the following objective:

$$\pi = \underset{(x,y_1,y_2) \in \mathcal{D}_{pool}}{\arg \max} \hat{U}(x, y_1, y_2) + \beta \hat{\mathcal{H}}(X_{tr} \cup \{x\}), \tag{5}$$

where $\hat{U}(x, y_1, y_2)$ is the preference model epistemic uncertainty estimate for the tuple $(x, y_1, y_2)$ and $\hat{\mathcal{H}}(X_{tr} \cup \{x\})$ is the entropy estimate *for the acquired prompt distribution*, assuming the policy selects $x$. $X_{tr}$ is a slight abuse of the notation that refers to the set of prompts in the previously acquired training set. Lastly, $\beta$ is a hyperparameter to balance the contribution of each term. Crucially, the first term represents a *task-dependent source* of epistemic uncertainty, since it refers to the learned preference Model. In contrast, the second term represents a *task-agnostic source*, as it solely relies on the information available in the feature space spanned by the base LLM.

**Preference Model Epistemic Uncertainty Estimation**. We first describe our Bayesian Preference Model. Assuming a prior distribution over parameters $p(\boldsymbol{\theta})$, the posterior predictive distribution over preferences after observing $\mathcal{D}_{train}$ is given by:

$$p(y_1 \succ y_2 \mid x, y_1, y_2, \mathcal{D}_{train}) = \int p(y_1 \succ y_2 \mid x, y_1, y_2, \boldsymbol{\theta}) p(\boldsymbol{\theta} \mid \mathcal{D}_{train}) d\boldsymbol{\theta}, \qquad (6)$$

where the likelihood term $p(y_1 \succ y_2 \mid x, y_1, y_2, \boldsymbol{\theta})$ follows the BT model in Equation 1. Considering deep models, solving this inference problem is intractable, given the large parameter space. Nonetheless, we may assume a simple yet effective posterior approximation via deep ensembles [49–51]:

$$p(\boldsymbol{\theta} \mid \mathcal{D}_{train}) \approx \sum_{k=0}^{K} \delta(\boldsymbol{\theta} - \hat{\boldsymbol{\theta}}_k). \qquad (7)$$

Equation 8 approximates the posterior distribution over parameters $p(\boldsymbol{\theta} \mid \mathcal{D}_{train})$ as a mixture of delta functions, where $K$ is the number of ensemble models and $\hat{\boldsymbol{\theta}}_k$ is the MAP estimate of model $k$. The posterior predictive distribution is then computed via the following approximation:

$$p(y_1 \succ y_2 \mid x, y_1, y_2, \mathcal{D}_{train}) \approx \frac{1}{K} \sum_{k=0}^{K} p(y_1 \succ y_2 \mid x, y_1, y_2, \boldsymbol{\theta}_k), \boldsymbol{\theta}_k \sim p(\boldsymbol{\theta} \mid \mathcal{D}_{train}). \qquad (8)$$

Equations 7 and 8 allow approximate inference by training multiple preference models separately. However, this is challenging in the context of LLMs, as fine-tuning billions of parameters several times is computationally expensive and impractical in many settings. Alternatively, we employ an ensemble of adapters [52, 53], which consists of multiple lightweight networks (with a few million parameters each) on top of the frozen LLM that works as a feature extractor. This allows us to generate the LLM features offline and use them as a dataset, considerably reducing the resources required for training and Bayesian inference. This also enables using very large base models, with dozens or hundreds of billions of parameters in a single GPU setting. Finally, based on the previous modeling assumptions, we can estimate the epistemic uncertainty term employing Equation 3:

$$\hat{U}(x, y_1, y_2) = \mathcal{H}(\frac{1}{K} \sum_{k=0}^{K} p(y_1 \succ y_2 \mid x, y_1, y_2, \boldsymbol{\theta}_k)) - \frac{1}{K} \sum_{k=0}^{K} \mathcal{H}(p(y_1 \succ y_2 \mid x, y_1, y_2, \boldsymbol{\theta}_k)). \qquad (9)$$

**Feature Space Entropy Estimation**. Equation 5 requires estimating the entropy of the acquired prompt distribution, $\mathcal{H}(X_{train})$. For this matter, we employ a kNN-based entropy estimator. We represent each prompt in the pool as the last-layer embedding vector generated by the base LLM, leveraging the semantic representations learned during unsupervised pre-training.

However, naively applying the KL estimator from Equation 4 has a major drawback: the training set $\mathcal{D}_{train}$ initially contains very few data points and does not provide support to represent the probability density, introducing bias to the estimates and affecting the data selection.

For illustration, we show the scenario of Figure 3a. In this case, we estimate the entropy using Equation 4, with $k = 3$. Since it does not account for the available points in the pool, it underestimates the density around the top cluster and ends up selecting the green point as the one that maximizes the entropy of the feature space, while the point that does so is in the bottom cluster. In an extreme case where all the points in

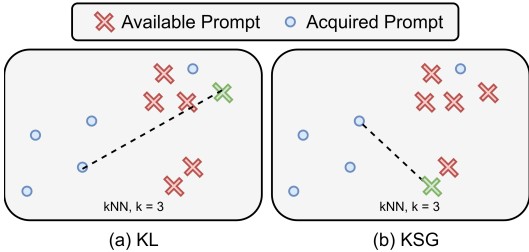

Figure 3: **Illustration of entropy estimators.** The green point maximizes the entropy estimation of the prompt distribution (according to the employed estimator). Dashed lines represent its k-NN distance. In (a), the KL estimator (Equation 4) does not account for the available prompts in the pool (in red) and underestimates the density in regions not covered by the acquired set (in blue). In (b), the KSG estimator (Equation 10) uses all data points, leading to better estimation and effectively selecting the point that maximizes the true entropy.

---

**Algorithm 1** BAL-PM

---

**Require:** Pool set $\mathcal{D}_{pool} = \{x^i, y_1^i, y_2^i\}^N$, training set $\mathcal{D}_{train} = \{x^i, y_1^i, y_2^i, y_1 \succ y_2{}^i\}^B$
**Require:** Base LLM $\tau$, Bayesian Preference Model $p(y_1 \succ y_2 \mid x, y_1, y_2)$
   Compute feature sets for $\mathcal{D}_{pool}$ and $\mathcal{D}_{train}$ by performing forward passes on $\tau$
   Compute kNN distances for points in $\mathcal{D}_{pool} \cup \mathcal{D}_{train}$
   **while** true **do**
      Train Bayesian Preference Model (ensemble) in $\mathcal{D}_{train}$ assuming Equations 7 and 8
      Compute Epistemic Uncertainty Estimates $\hat{U}(x, y_1, y_2)$ via Equation 9
      Initialize $n_{X_{tr}}(x)$ by counting $\{u \mid u \in \mathcal{D}_{train} \wedge (\|x - u\| \leq D(x)/2\}, \forall x \in \mathcal{D}_{pool}$
      Initialize Batch: $\mathcal{B} = \{\}$
      **while** batch not full **do**
         Compute entropy term: $e(x) = \log D(x) - \frac{1}{d_X}\psi(n_{X_{tr}}(x) + 1)$
         Select tuple $(x^*, y_1^*, y_2^*)$ following $\pi = \underset{(x,y_1,y_2)\in\mathcal{D}_{pool}}{\arg\max}\ \hat{U}(x, y_1, y_2) + \beta e(x)$
         Update Pool and Batch: $\mathcal{D}_{pool} = \mathcal{D}_{pool} \backslash (x^*, y_1^*, y_2^*)$, $\mathcal{B} = \mathcal{B} \cup (x^*, y_1^*, y_2^*)$
         Update counts: $\forall x \in \mathcal{D}_{pool}, n_{X_{tr}}(x) = n_{X_{tr}}(x) + 1$ if $\|x - x^*\| \leq D(x)/2$
      **end while**
      Collect human feedback for $\mathcal{B}$ and update training set $\mathcal{D}_{train} = \mathcal{D}_{train} \cup \mathcal{B}$
   **end while**

---

the top cluster are the same, this bias leads to acquiring duplicated points. In Appendix E we formally derive the KL entropy estimator and show how the low-data regime challenges its main assumptions.

Alternatively, we may use the available unlabeled pool, often much larger than the acquired set. Following the argument introduced by Kraskov et al. [54], the key insight is to notice that Equation 4 holds for *any* value of $k$ and it does not require a fixed $k$ over different particles for entropy estimation (we provide more details in Appendix E). Therefore, we can find the distance to the $k$-th nearest neighbor in the joint space spanned by the pool and the acquired set and map it to the corresponding neighbor (denoted as $n_{X_{tr}}$) in $X_{train}$ to estimate the marginal entropy. This results in the KSG marginal entropy estimator [54], but repurposed to our setting:

$$\hat{\mathcal{H}}_{KSG}(X) = \frac{d_X}{N}\sum_{i=0}^{N}\log D_x(i) + \log v_{d_X} + \psi(N) - \frac{1}{N}\sum_{i=0}^{N}\psi(n_{X_{tr}}(i) + 1), \qquad (10)$$

where $D_x(i)$ is now computed in the joint space and $n_{X_{tr}}(i)$ is the number of points in $\mathcal{D}_{train}$ whose distance to $x_i$ is less than $D_x(i)/2$. Figure 3 (b) illustrates the data selection by following this alternative estimation, leading to more diversity in the feature space.

**Implementation**. Firstly, we simplify the entropy term by dropping the constant terms with respect to $x$:

$$\underset{(x,y_1,y_2)\in\mathcal{D}_{pool}}{\arg\max}\ \hat{\mathcal{H}}(X_t \cup \{x\}) = \underset{(x,y_1,y_2)\in\mathcal{D}_{pool}}{\arg\max}\ \log D(x) - \frac{1}{d_X}\psi(n_{X_{tr}}(x) + 1). \qquad (11)$$

Equation 11 acquire points by computing $D(x)$ (based on the kNN distance) and the counter $n_{X_{tr}}$ related to prompt $x$ only. Furthermore, as $D(x)$ accounts for the full dataset in the KSG estimator, it does not change over training. Hence, we may compute it offline once, and potentially scale to very large datasets [55]. Lastly, BAL-PM acquisition scheme builds a batch of data by successively selecting points following Equation 5. Crucially, while BAL-PM keeps the preference model uncertainty estimates the same over the batch, it updates the entropy term after in-batch iteration. This operation boils down to updating the counter $n_{X_{tr}}$, a lightweight operation. In Algorithm 1, we present the pseudocode for BAL-PM. Appendix H further describes its computational cost.

## 5 Experiments and Discussion

In this Section, we aim to evaluate how BAL-PM performs in Active Preference Modeling. Our central hypothesis is that leveraging the information available on the feature space spanned by the

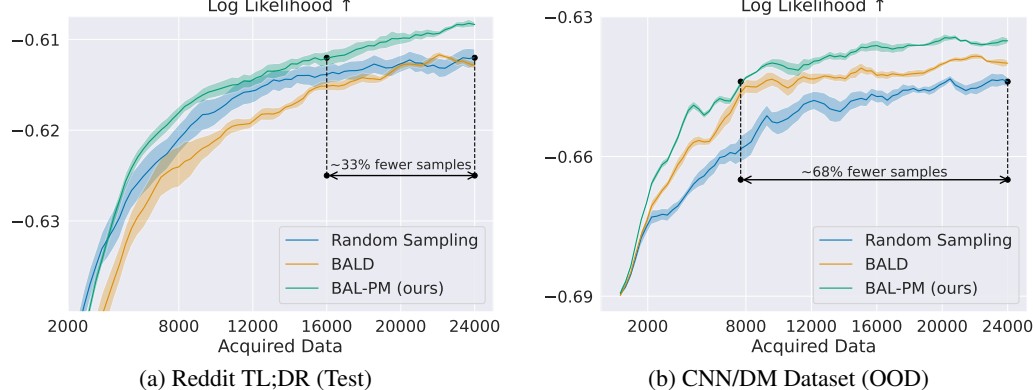

(a) Reddit TL;DR (Test)    (b) CNN/DM Dataset (OOD)

Figure 4: **Comparison with baseline methods in Active Preference Modeling**. BAL-PM considerably reduces the number of samples required for preference modeling, achieving **33%** and **68%** of reduction in the Reddit TL;DR test split and CNN/DM News datasets, respectively. The shaded area corresponds to the standard error computed over five seeds.

base LLM — a task-agnostic source of epistemic uncertainty – addresses the problem of acquiring redundant samples, a natural pathology of relying on task-dependent epistemic uncertainty estimators designed for single-point acquisition schemes. BAL-PM, our proposed stochastic acquisition policy, promotes this diversity by maximizing the entropy of the acquired prompt distribution, besides selecting points for which the preference model presents high epistemic uncertainty.

**Experimental Setup**. We consider a pool-based active learning setup. Each experiment consists of several acquisition cycles, where each iteration performs a batch acquisition in the currently available pool. The training set starts with the size of one acquired batch and leaves the remaining data for the pool set. Following previous works [13, 14], we reinitialize the model after each acquisition to decorrelate subsequent acquisitions. We train the ensemble of adapters on previously acquired data and employ early stopping based on the Log-Likelihood of a held-out validation set. We evaluate the preference model after each acquisition loop and report the average Log-Likelihood of the test sets. Appendix G discusses why the test average Log-Likelihood is a proper metric for Preference Modeling. Finally, Appendix C details hyperparameters and tuning methodology used in this work[4].

**Model Architecture**. As described in Section 4, we employ an ensemble of adapters on top of a base LLM. Each adapter is a multi-layer perceptron with non-linear activations. In most experiments, the base LLM is a 7-billion parameter model, although we also employed 70-billion and 140-billion parameter ones when analyzing the effect of scaling the base LLM. All considered models are only unsupervised pre-trained and have not undergone any preference fine-tuning.

**Datasets**. Following previous work [20], we considered prompts from the Reddit TL;DR dataset of Reddit posts [25] and the CNN/DM News dataset [26]. We leverage the generated completions and human feedback collected by Stiennon et al. [1]. The Reddit dataset contains train/eval/test splits, and we adopt the train split (92,858 points) for the pool and training sets, the eval split (33,083 points) for validation, and report results in the test set (50,719 points). The CNN/DM dataset contains a single split (2,284 points), and we use it for the Out-Of-Distribution (OOD) evaluation.

**Comparison Methods**. We considered **Random Sampling** and **BALD** [23] as baselines. BALD selects points based on the utility function of Equation 3 and is equivalent to the acquisition function used by Gleave and Irving [20]. We also compared BAL-PM with other stochastic acquisition policies [22], namely **SoftmaxBALD**, **SoftRankBALD**, and **PowerBALD**. We refer to Kirsch et al. [22] for a detailed description of these methods.

## 5.1 Experiments

We highlight and analyze the following questions to evaluate our hypothesis and proposed method.

---

[4]We release our code at `https://github.com/luckeciano/BAL-PM`.

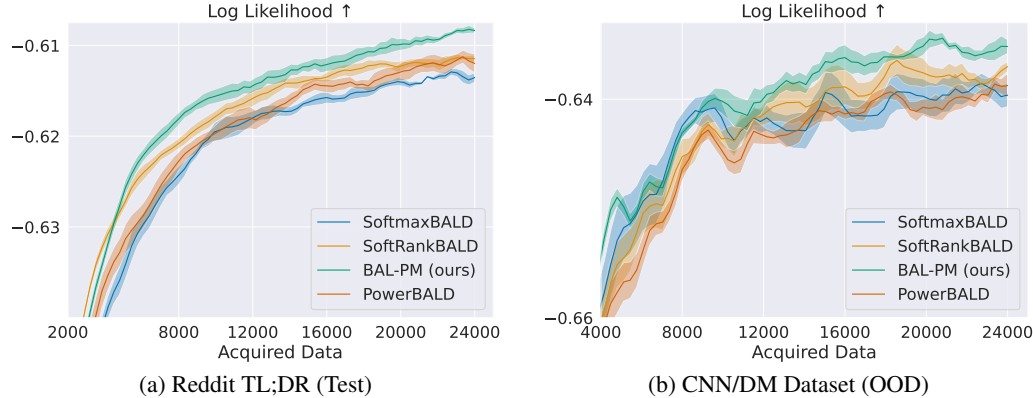

(a) Reddit TL;DR (Test)          (b) CNN/DM Dataset (OOD)

Figure 5: **Comparison with Bayesian stochastic acquisition policies for Active Preference Modeling**. BAL-PM consistently outperforms other policies in Test and OOD settings.

**Does BAL-PM reduce the volume of feedback required for Preference Modeling?** We start evaluating how BAL-PM performs against standard random acquisition and BALD, as presented in Figure 4. BAL-PM considerably reduces the volume of data required to learn the preference model. Particularly compared with random sampling, it reduces the number of required samples in **33%** for the Reddit TL;DR dataset and **68%** for the out-of-distribution setting of CNN/DM News dataset, representing a substantial reduction in the human feedback needed. BALD does not present any benefits over random sampling in the TL;DR dataset, which aligns with previous work [20]. Interestingly, BALD also presents an interesting improvement over random sampling in the OOD setting, but BAL-PM consistently outperforms BALD with more data.

**How does BAL-PM compare with other stochastic acquisition policies?** Next, we analyze BAL-PM in comparison with other Bayesian stochastic acquisition policies. These policies address the acquisition of redundant samples by relying on sampling points from the distribution determined by the task-dependent epistemic uncertainty scores. BAL-PM consistently surpasses all variations in both datasets, suggesting that leveraging the information available in the prompt feature space – as a task-agnostic source of epistemic uncertainty – is more effective in encouraging diversity for batch acquisition in the considered setting.

**Does BAL-PM encourage diversity and prevent the acquisition of redundant samples?** We evaluate the exploration approach of the considered methods by analyzing the statistics of the acquired prompt distribution, particularly the number of unique prompts over the course of training.

Figure 6 presents three different perspectives on the acquired distribution. On the left, it presents the number of unique acquired prompts over learning, which indicates diversity in the training set. BAL-PM selects new prompts at a much faster rate than random sampling and BALD. Naturally, this rate saturates when the selection exhausts the number of distinct prompts available in the pool (approximately 14,000). The rate is also not equivalent to the data acquisition rate since BAL-PM does not simply select different prompts but also prioritizes points with high epistemic uncertainty.

The middle plot shows the ratio of unique prompts in each active learning loop, and BAL-PM acquires batches with all distinct prompts during almost the whole training. BALD only maintains a rate of 70%, which means a substantial number of duplicated prompts. In Appendix K, we present the first batch sampled by BALD and BAL-PM for a qualitative analysis. Lastly, the plot on the right shows the ratio of unique prompts across all training. While random sampling presents a high unique prompt ratio in each separate batch, it consistently samples duplicated prompts throughout learning. In contrast, BAL-PM maintains a high ratio of unique prompts during most of the training. Again, this rate progressively decays as BAL-PM exhausts the pool of different prompts and due to the influence of the epistemic uncertainty prioritizing particular prompt-completion pairs.

**How does BAL-PM scale to larger LLMs?** As highlighted in Section 4 our design choices allow us to scale our experiment for very large base LLMs in a single GPU setting. We investigate the effect of scaling the base LLM in BAL-PM performance, considering 70-billion and 140-billion parameter models in their 4-bit quantized versions. Naturally, the preference model performance improves substantially against the 7-billion parameter model. More interestingly, BAL-PM presents similar

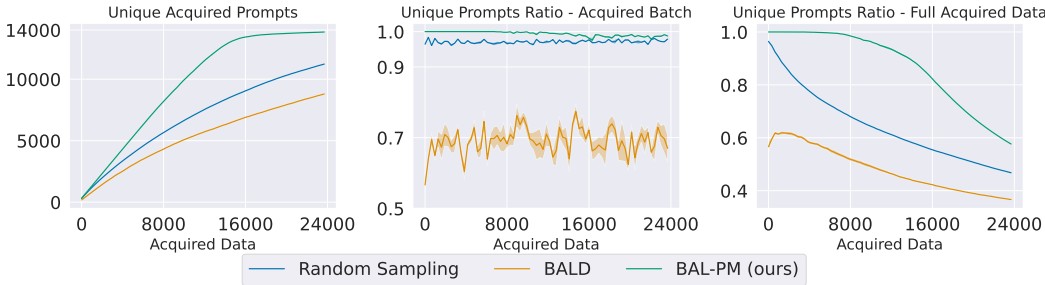

Figure 6: **Statistics of acquired prompt distribution**. We present the total number of unique acquired prompts (left), the ratio of unique acquired prompts per active learning loop (middle), and the ratio of unique acquired prompts over training. BAL-PM consistently acquires novel prompts and encourages diversity in each acquired batch and the full training set.

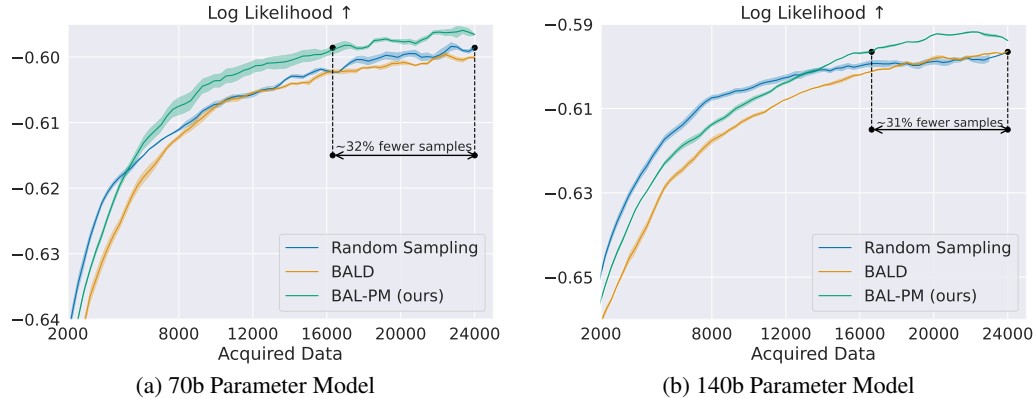

Figure 7: **The effect of scaling the base LLM**. We analyzed how increasing the size of the base LLM affects BAL-PM performance in the Reddit TL;DR dataset. We considered (a) a 70-billion parameter model and (b) a 140-billion parameter model. Interestingly, we find approximately the same gains (31%–33% reduction of required samples) across all models.

gains across all scales, with around 31%–33% reduction of required samples compared to random sampling. In contrast, BALD still does not present benefits over random sampling, suggesting that the scale of the base LLM is not the prevailing factor for its negative result.

**Ablations and Further Analysis.** We conduct ablation studies in the key components of the proposed method in Appendix D. More concretely, we ablate the components of the objective to show that both preference model epistemic uncertainty and entropy scores play a relevant role in BAL-PM. We also ablate the type of uncertainty and the employed entropy estimator. Furthermore, we conduct further empirical analysis in Appendix F to investigate how each component of Equation 5 contributes to the data selection, and conduct a robustness analysis for the $\beta$ hyperparameter in Appendix I. Lastly, we provide comparison with additional data selection baselines in Appendix J.

## 6 Closing Remarks

In this work, we present BAL-PM, a Bayesian Active Learning method for Preference Modeling in Language Models. BAL-PM is a stochastic acquisition policy that selects points for which the preference model presents high epistemic uncertainty and also maximizes the entropy of the acquired prompt distribution. We show that leveraging the information available on the feature space spanned by the base LLM via this entropy term has a crucial role in preventing the acquisition of redundant samples. BAL-PM substantially reduces the volume of feedback required for Preference Modeling and outperforms existing Bayesian stochastic acquisition policies. It also scales for very large LLMs and effectively balances the contribution of both considered sources of uncertainty.

**Limitations**. Despite its encouraging results, BAL-PM presents some limitations. For instance, it heavily relies on the quality of the feature representations provided by the base LLM. Particularly, it might be subject to the Noisy-TV problem [56] and provide high-entropy scores to nonsensical prompts if those are spread in the representation space rather than collapsed into a single region. Fortunately, we expect this limitation to be progressively addressed by better LLMs.

**Future Work** may evaluate BAL-PM in larger preference datasets with millions or billions of data points. Another direction analyzes how the learned models perform in the Preference Optimization setting. Lastly, future work may extend BAL-PM to consider recent prediction-oriented methods of epistemic uncertainty estimation [57] in contrast to parameter-based methods such as BALD.

## Acknowledgments and Disclosure of Funding

We thank Jannik Kossen for the insightful discussions in the early stages of this project. We also thank the reviewers for providing insightful feedback. Luckeciano C. Melo acknowledges funding from the Air Force Office of Scientific Research (AFOSR) European Office of Aerospace Research & Development (EOARD) under grant number FA8655-21-1-7017.

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

# A    Impact Statement

Preference fine-tuning has become a crucial step in aligning LLMs toward human preferences and demonstrated a real-world impact in many open-source and production systems [2, 10]. Nonetheless, collecting human feedback is very expensive and time-consuming, posing a substantial bottleneck for further development of LLMs. In this work, we approach the problem of Active Preference Modeling, which aims to reduce the volume of feedback required for learning preferences. We show that our proposed method, BAL-PM, requires 33% to 68% fewer labels in the human preference datasets considered. We believe that these results point out to **a strong impact in the process of acquiring labels**, and estimate an **economy of hundreds of thousands of dollars and months of labeling work in the current scale of LLMs**. This scenario represents faster cycles of preference optimization, potentially leading to better-aligned and safer models. Therefore, we believe our work poses a relevant positive societal impact for the upcoming years.

# B    Reproducibility Statement

**Code Release.** To ensure the reproducibility of our research findings, we release our code at `https://github.com/luckeciano/BAL-PM`. Our implementation is based on PyTorch [58] and HuggingFace [59]. All baselines are available in the released code.

**Experiments Reproducibility.** We detail our methodology in Section 4 and our experimental setup in Section 5. We provide all hyperparameters used in this work as well as the selection strategy in Appendix C. We plan to release all the raw experiment logs and feature datasets generated in this work. For all experiments in this paper, we report the results over five seeds with standard errors. For better visualization, we applied smoothing for the curves considering two past observations.

**Datasets.** All preference datasets are open-source and available online for academic use [1].

**Compute Resources.** We execute all active learning experiments in a single A100 GPU, and each experiment takes approximately one day. For the base LLM feature generation, we also use a single A100 GPU, taking a few hours for the 7-billion parameter model and approximately four days for the 70-billion and 140-billion parameter models.

## C  Hyperparameters

In Table 2, we share all hyperparameters used in this work. We specifically performed a hyperparameter search on the entropy term parameters and baselines. The search strategy was a simple linear search on the options in Table 1, considering each parameter in isolation. The selection followed the final performance on a held-out validation set. For baselines, we mostly considered the values presented in prior work [22]. For the proposed method, we also considered $d_X$ as a hyperparameter and found that smaller values often work better than using the dimensionality of the base LLM embeddings.

| Hyperparameter | Value |
|---|---|
| Acquisition Batch Size | 320 |
| Initial Training Set Size | 320 |
| Initial Pool Size | 92,000 |
| Active Learning Iterations | 75 |
| Adapter Network Hidden Layers | [2048, 256] |
| Adapter Network Activation Function | tanh |
| 7b Model Name | OpenHermes-2.5-Mistral-7B |
| 70b Model Name | Llama3-70b |
| 140b Model Name | Mixtral-8x22B-v0.1 |
| Early Stopping Patience | 3 |
| Training Batch Size | 32 |
| Learning Rate | 3e-5 |
| Learning Rate Scheduler | Cosine |
| Optimizer | AdamW |
| Entropy Term $\beta$ | 0.01 |
| Entropy Term $k$ | 13 |
| Entropy Term $d_X$ | 1.0 |
| SoftmaxBALD $\beta$ | 10,000 |
| SoftRankBALD $\beta$ | 1.0 |
| PowerBALD $\beta$ | 8.0 |

Table 1: **Training Hyperparameters**.

| Hyperparameter | Search Space |
|---|---|
| Entropy Term $\beta$ | [0.0001, 0.001, 0.01, 0.1, 1.0] |
| Entropy Term $k$ | [1, 7, 13, 19, 25] |
| Entropy Term $d_X$ | [4096, 2048, 1024, 256, 64, 32, 8, 4, 2, 1, 0.5] |
| SoftmaxBALD $\beta$ | [0.25, 1.0, 2.0, 100.0, 1000.0, 5000.0, 10,000] |
| SoftRankBALD $\beta$ | [0.25, 1.0, 2.0, 4.0, 8.0] |
| PowerBALD $\beta$ | 0.25, 1.0, 2.0, 4.0, 8.0, 10.0, 12.0 |

Table 2: **Hyperparameters search space**.

# D   Ablation Studies

This Section presents and discusses the results of the ablation studies. We focused on three different aspects: the components in the objective of Equation 5; the nature of the uncertainty considered; and the entropy estimator.

**Objective Components.** We considered three different versions for ablating components: BAL-PM (ours), which follows Equation 5 exactly; a version with **No Uncertainty Score** in the objective; and another version with **No Entropy Score**. Figure 8 shows the findings. In the datasets considered, both terms of the objective play a crucial role in the performance of BAL-PM. Disregarding the entropy score fundamentally means solely following BALD, which acquires several redundant samples. On the other side, disregarding the uncertainty score prevents the learner from acquiring points where the model lacks information.

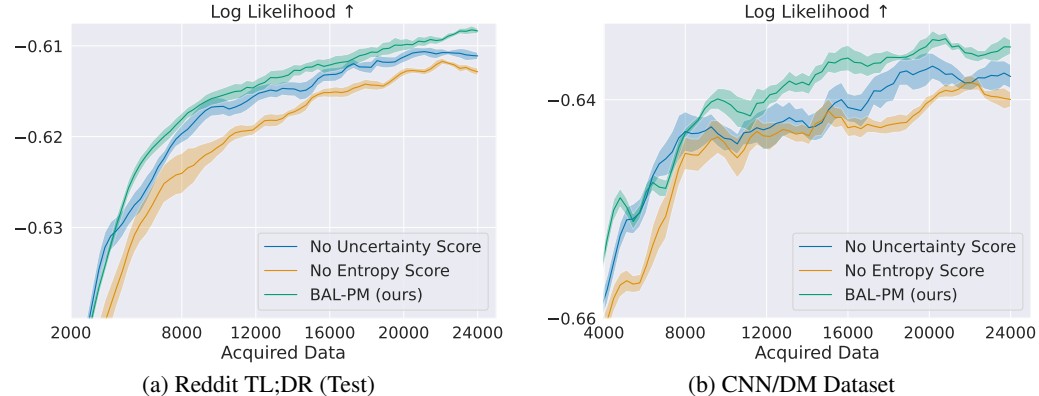

(a) Reddit TL;DR (Test)                              (b) CNN/DM Dataset

Figure 8: **The ablation study of the components in the BAL-PM objective.** We considered BAL-PM, a version without the uncertainty score, and a version without the entropy score.

**Type of Uncertainty**.  In Machine Learning, we identify two different sources of uncertainty: *epistemic* and *aleatoric*. Epistemic Uncertainty refers to the uncertainty in the parameters of the model, often due to the lack of information from some regions of the parameter space. In contrast, aleatoric uncertainty refers to the uncertainty in the *data*, originating from the inherent noise of the data generation process. We reduce epistemic uncertainty by acquiring new data, while aleatoric is irreducible.

A common practice in Active Learning is to select points based on high *Predictive Uncertainty*, which is often referred to as "Uncertainty Sampling" [60]. This type represents the total uncertainty, i.e., it accounts for both epistemic and aleatoric sources. Therefore, we expect that following Predictive Uncertainty underperforms in datasets with high label noise, as the objective may favor points with high aleatoric uncertainty and low epistemic uncertainty.

Figure 9 compares using **Predictive** and **Epistemic** uncertainties in the objective of Equation 5. Selecting points based on epistemic uncertainty strongly outperforms the other variant, which aligns with the fact that preference datasets contain high levels of label noise – as mentioned in Section 1, the agreement rate among labelers is low, typically between 60% and 75%. This ablation also highlights the importance of a Bayesian preference model for epistemic uncertainty estimation.

**Entropy Estimator.** In Section 5, we argue for using the KSG entropy estimator (rather than the KL estimator) since it leverages the full dataset and better estimates the probability density in the feature space, leading to less biased entropy estimates. In this ablation, we compare both estimators to measure the impact of this design choice.

Figure 10 presents the results of this ablation. In the Reddit TL;DR dataset, the KSG estimator consistently outperforms the KL estimator, requiring approximately 20% fewer samples. In the OOD setting, both estimators performed equally. This is expected once that the available pool and training set does not provide support in the regions of the feature space with out-of-distribution samples.

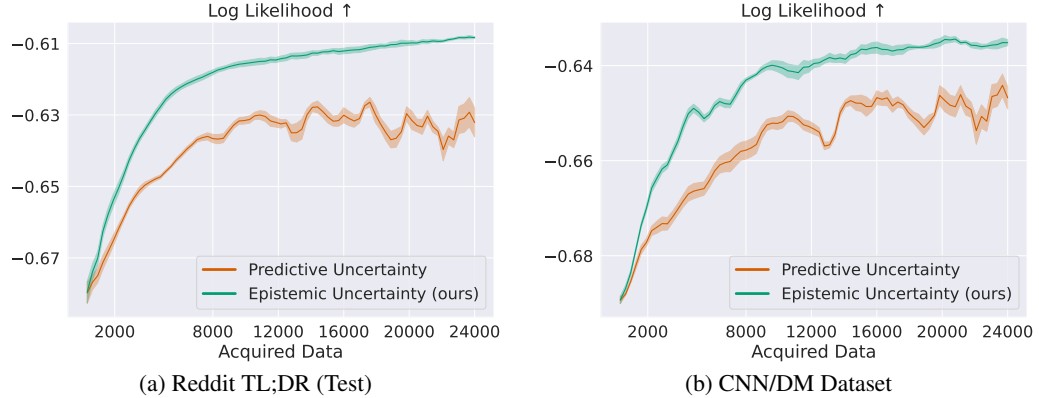

(a) Reddit TL;DR (Test)        (b) CNN/DM Dataset

Figure 9: **The ablation study of the type of Uncertainty in the BAL-PM objective**. Leveraging Epistemic Uncertainty substantially surpasses Predictive Uncertainty since it disregards the effect of the high Aleatoric Uncertainty from preference datasets.

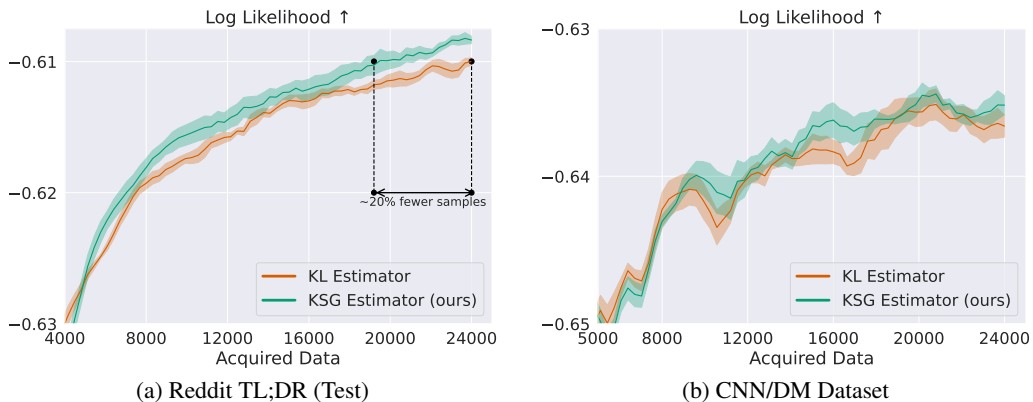

(a) Reddit TL;DR (Test)        (b) CNN/DM Dataset

Figure 10: **The ablation study of the type of entropy estimator in the BAL-PM objective**. Using the KSG estimator requires approximately 20% fewer samples than the KL estimator in the Reddit TL;DR dataset.

# E   KL Entropy Estimator: Review and Assumptions

In this Section, we review the derivation of the KL entropy estimator and highlight the main assumptions and how they impacted the design choices for BAL-PM. We mostly follow the derivation from Kraskov et al. [54].

We start defining $X$ as a continuous random variable in a vector space where the Euclidean norm $\|x - x^*\|$ is well-defined ($x$ and $x*$ are two realizations of $X$). Let $\mu(x)$ represent the density of $X$ over this vector space. The Shannon entropy is defined as:

$$\mathcal{H}(X) := -\mathbb{E}_{\mu(x)}[\log \mu(x)] = -\int \mu(x) \log \mu(x) dx. \tag{12}$$

To build an estimator, we can approximate Equation 12 via Monte-Carlo sampling:

$$\hat{\mathcal{H}}(X) = \frac{1}{N} \sum_{i=0}^{N} \log \hat{\mu}(x)), \tag{13}$$

where $N$ is the number of samples for approximation and $\hat{\mu}(x)$ is an estimate of the density of $X$.

The goal of kNN-based entropy estimators is primarily to provide a good approximation for the density. For this matter, we first define a probability distribution $P_k(\epsilon)$ for the distance between any realization $x_i$ and its k-nearest Neighbor. We start highlighting the first assumption:

**Assumption 1.** *The probability $P_k(\epsilon)d\epsilon$ is equal to the chance of existing one point such that $\|x - x^*\| < \epsilon/2$, $k - 1$ other points with smaller distances, and $N - k - 1$ points with larger distances.*

Following this assumption we can obtain the following trinomial distribution:

$$P_k(\epsilon) = k \binom{N-1}{k} \frac{dp_i(\epsilon)}{d\epsilon} p_i^{k-1} (1 - p_i)^{N-k-1}, \tag{14}$$

where $p_i(\epsilon)$ is the probability mass of the $\epsilon$-ball centered in $x_i$. The expectation of $\log p_i(\epsilon)$ is:

$$\mathbb{E}[\log p_i(\epsilon)] = \int_0^\infty P_k(\epsilon) \log p_i(\epsilon) d\epsilon = k \binom{N-1}{k} \int_0^1 p_i^{k-1} (1 - p_i)^{N-k-1} \log p_i$$
$$= \psi(N) - \psi(k), \tag{15}$$

where $\psi$ denotes the digamma function. We now highlight the second assumption:

**Assumption 2.** *The density $\mu(x)$ is constant in the $\epsilon$-ball.*

Assumption 2 allows us to approximate $p_i(\epsilon) \approx c_d \epsilon^d \mu(x_i)$, where $d$ is the dimension of $x$ and $c_d$ is the volume of the $d$-dimensional unit ball. Using Equation 15 in this approximation and rearranging terms, we have:

$$\log \hat{\mu}(x_i) \approx \psi(k) - \psi(N) - d\mathbb{E}[\log \epsilon] - \log c_d. \tag{16}$$

Finally, using this estimator in Equation 13, we obtain the KL entropy estimator in Equation 4.

**Remarks**. Now, we analyze how this derivation and assumptions impact our entropy estimator. First, Assumption 1 models the probability based on the choice of $k$. For the low-data regime (i.e., $N$ is small), this could lead to considerably large $\epsilon$-balls where the Assumption 2 does not hold, and, therefore, it is not a good approximation. Thus, naively applying the KL estimator in the acquired training set may lead to strongly biased entropy estimates.

Secondly, in Section 4, we raise the key insight that Equation 4 holds for *any value of $k$*, and it does not require a fixed $k$ over different particles for entropy estimation. Indeed, the density estimation $\hat{\mu}(x_i)$ is estimated for each particle $x_i$ in isolation (Equation 16). Therefore, we may choose a different $k$ for each particle to ensure that Assumptions 1 and 2 are valid.

## F    BAL-PM Objective – Empirical Analysis

**Balancing Task-Dependent and Task-Agnostic Epistemic Uncertainty for Active Learning**. Since considering the information in the feature space is crucial for Active Preference Modeling, a relevant question arises: how should an algorithm balance the contributions between the Bayesian preference model epistemic uncertainty and the prompt feature space uncertainty? Excessive reliance on the task-dependent term leads to acquiring redundant points. Similarly, the exacerbated contribution of the task-agnostic term prevents the acquisition of the points that reduce the uncertainty in the preference model. Thus, we investigate how BAL-PM balances these two terms over active learning. In Figure 11, we show the ratio of the entropy and preference model epistemic uncertainty scores in the first selected point of each acquired batch. Interestingly, BAL-PM automatically adjusts the contribution of each term. It progressively decays and converges the influence of the entropy score (task-agnostic source) as the novelty in the prompt feature space reduces due to the acquisition of new points. Similarly, it increases the relevance of the preference model uncertainty estimates (task-dependent source). A positive downstream effect is that BAL-PM switches to a more task-dependent selection as it improves the Bayesian model and, consequently, its epistemic uncertainty estimates.

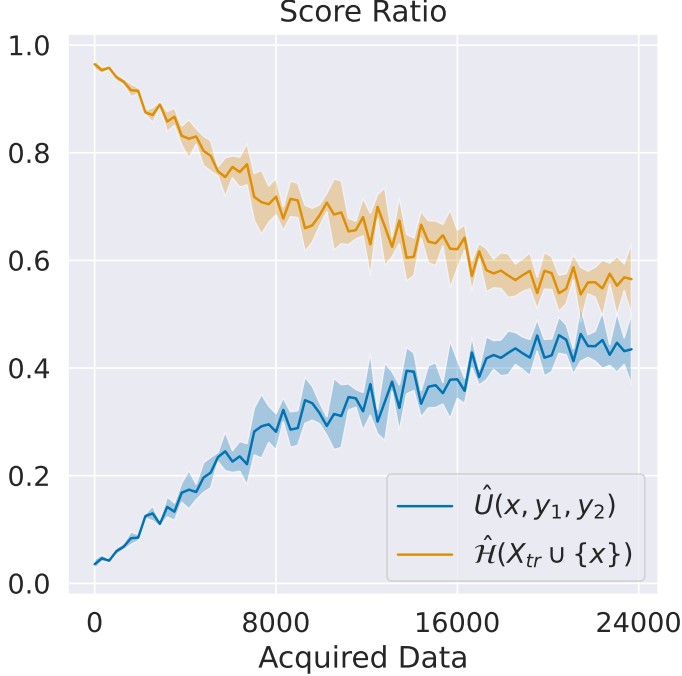

Figure 11: **Ratio of entropy and preference model uncertainty scores**. This plot represents the normalized contributions from the terms of Equation 5 on the first selected point of each batch. BAL-PM automatically adjusts the contribution based on the information available in the pool set.

# G   Is Log-Likelihood a proper performance measure for Preference Modeling?

In this Section, we argue why the Average Log-Likelihood on the test set is a good performance measure for Preference Modeling. Given a test set $\mathcal{D}_{test} = \{(x, y_1, y_2, y_1 \succ y_2)\}^N$ and the learned preference model $p_{\boldsymbol{\theta}}(y_1 \succ y_2 \mid x, y_1, y_2)$, the average Log-Likelihood is given by:

$$LL(\mathcal{D}_{test}, \boldsymbol{\theta}) = \mathbb{E}_{(x, y_1, y_2, y_1 \succ y_2) \sim \mathcal{D}_{test}}[\log p_{\boldsymbol{\theta}}(y_1 \succ y_2 \mid x, y_1, y_2)]. \tag{17}$$

Equation 17 is exactly the objective maximized in standard binary classification (or, equally, the minimization of the negative Log-Likelihood loss) but computed over the test data. In other words, this is the negative "test loss".

Average LL is a typical metric in the Active Learning and Uncertainty Quantification literature [61–63]. For Preference Modeling, it is very relevant as **LL directly accounts for the *preference strength to rank models***: given a triple $(x, y_1, y_2)$ where all raters agree that $y_1$ is preferable over $y_2$, LL allows us to measure that a model A predicting $p_A(y_1 \succ y_2 \mid x, y_1, y_2) = 0.9, (LL \approx -0.1)$ is better (in that data point) than another model B predicting $p_B(y_1 \succ y_2 \mid x, y_1, y_2) = 0.6$ $(LL \approx -0.5)$. Accuracy would provide an equal score for both models since it only accounts for the binarized prediction. LL provides a more "fine-grained" measure.

Another crucial point is that **LL factors in the aleatoric uncertainty in the label-generating process**. For instance, in a scenario where only 70% of the raters agree that $y_1$ is preferable, LL better ranks models whose predictions are closer to $p = 0.7$, respecting the ground truth preference strength, which is not possible with accuracy.

## G.1   Do the models better ranked by Average Log Likelihood (LL) lead to better fine-tuned policies?

In the context of Preference Modeling, fine-tuning LM policies is currently a very relevant downstream task. The Preference Modeling optimization objective and model selection protocol adopted in this work follow exactly the prior influential work on the topic [6, 1], which provides evidence that better preference models (in terms of validation loss) lead to improved downstream policies. Thus, we expect our models to behave similarly under the same conditions.

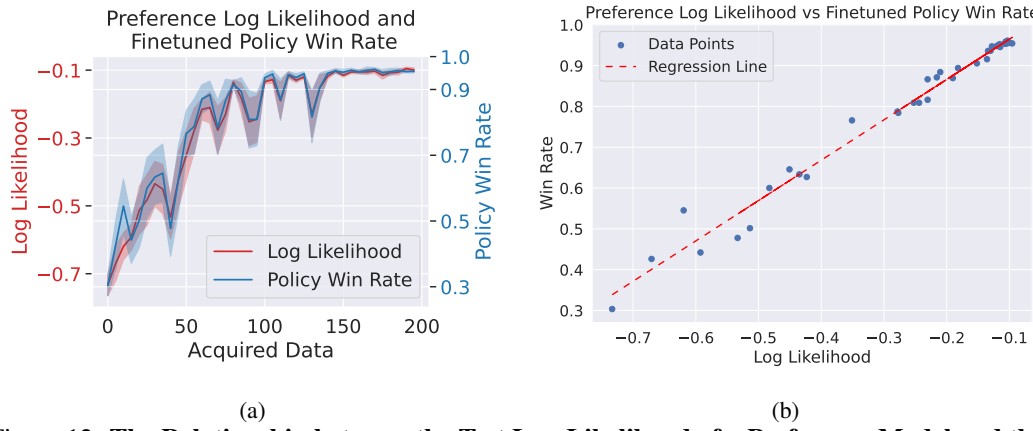

(a)                                                          (b)

Figure 12: **The Relationship between the Test Log-Likelihood of a Preference Model and the Performance of the corresponding fine-tuned Policy**. We show that, under simple conditions, there is a strong correlation between these two performance measures.

As additional evidence, we empirically illustrate the relationship between Log-Likelihood and policy performance on a simplified setup (Figure 12). Here, prompts $x$ and completions $y$ are real numbers in $[0, 1]$. The ground-truth reward function is given by a Gaussian density function $r(x, y) = \mathcal{N}(x + y \mid \mu = 1.0, \sigma = 0.4)$, and true preferences follow the Bradley-Terry model. In this setup, we progressively increase the training set size (the x-axis in Figure 12a) at which we train the preference models. This process generates different models with increasing levels of

test-set average Log-Likelihood. Then, similar to [64], we optimize the base policy via a Best-of-N optimizer by leveraging each of these learned preference models. Finally, we report the rate at which the fine-tuned policy completion is preferable over the base policy completion according to the ground truth reward model ("win rate"). Although simple, this setting allows us to bypass several optimization and distributional challenges and solely focus on evaluating the relationship between average Log-Likelihood and the performance of the fine-tuned policy. Figure 12a reports the Log-Likelihood (red) and the win rate against the base policy (blue). Figure 12b directly plots both measures and fits a regression line. We observe a strong correlation, which aligns with our point: **a higher test-set average Log-Likelihood means that the preference model is better at predicting the ground truth preferences, assigning higher rewards for better completions, and, therefore, improving fine-tuned policies that maximize such reward scores**.

## H  BAL-PM Computational Cost Description

In this section, we describe the computational cost of executing BAL-PM. We argue that computational tractability is one of the main contributions of our method. We start by providing some context: our work focuses on (Bayesian) Active Learning, which is naturally more computationally demanding than simply training predictive models. This is because **we require models that express epistemic uncertainty** to acquire informative labels for efficient training. This also **requires models to constantly update their uncertainties based on the new data via re-training**. The key is that **Active Learning reduces the number of labels needed to train a better model, which considerably overcomes the extra computational cost**. Labeling is significantly more expensive and laborious.

As described in Section 1, Preference Modeling in LLMs requires batch acquisition – it is impossible to request the label of a single point, re-train the model, and repeat this process. Still, tractable methods rely on these single-point acquisition objectives. Thus, **what BAL-PM does computationally is to replace $B$ model re-trainings per acquired batch with computing entropy estimates** (considerably cheaper, as explained below). $B$ is the batch size, and we set $B = 320$ in our experiments.

**BAL-PM does not require training or inference on LLMs during the active learning loops**. This considerably reduces the computational cost and allows us to scale up to 140b models in a single A100 GPU. Comparatively, even fully fine-tuning a 7b-parameter model currently requires at least several A100 GPUs. LoRA methods [65] also require new LLM inferences for every model update, while BAL-PM only requires a single time.

The computation of BAL-PM has three pieces: offline processing (LLM inference and kNN computation), updating adapters, and entropy estimation. LLM inference is done only once before Active Learning, which is the minimum for LLM adoption. Furthermore, **we may compute the features used for the preference model and sentropy estimation in the same forward pass**: every prompt-completion input concatenates prompt/completion texts. Thus, we can extract prompt features as the last layer embedding right after the last prompt token and the prompt-completion features right after the completion's last token. Hence, there is no extra cost to extract features for entropy estimation. The cost of updating adapters is minimal: it boils down to updating MLPs with two hidden layers, which is reasonably cheap for LLM research. Lastly, the entropy estimation only requires computing the di-gamma function (Equation 11) in the pool.

# I  $\beta$ Robustness Analysis

In this Section, we introduce a robustness analysis for the $\beta$ hyperparameter (Figure 8) considering the values in the search space, described in Table 2. As presented in Equation 5, this hyperparameter balances the effect of the epistemic uncertainty and entropy scores.

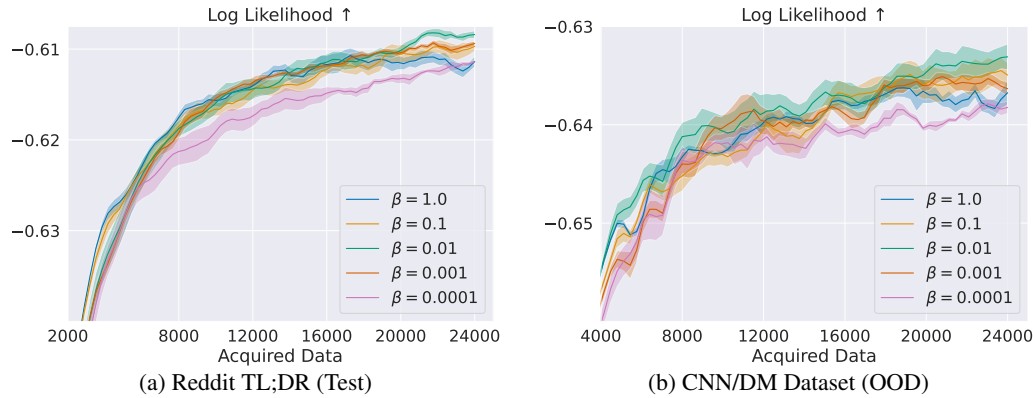

(a) Reddit TL;DR (Test)  (b) CNN/DM Dataset (OOD)

Figure 13: $\beta$ **Robustness Analysis**. We considered different scales of the $\beta$ hyperparameter in the BAL-PM objective (Equation 5).

The impact of the choice is more noticeable when values are 100x greater/lower than the optimal choice. Values around 10x greater/lower still perform well, suggesting good room for choosing this hyperparameter. Furthermore, we employed the same value of $\beta$ across the different datasets and LLMs, suggesting robustness across different relevant dimensions. Crucially, $\beta$ trades off the contribution of two different terms. As such, it provides a spectrum of objectives and may recover the two extremes presented in the ablation of Figure 13. Naturally, different choices of $\beta$ will change the uncertainty score ratio in Figure 11 on Appendix F (i.e., the contribution of each term after convergence). Nevertheless, and most importantly, the behavior of the curves — the entropy contribution progressively reducing and converging and the relevance of epistemic uncertainty estimates increasing — should remain.

## J    Further Baselines

In this section, we provide additional baselines for further comparison.

### J.1    Full Dataset Baseline

We start by evaluating the performance of a preference model trained in the full dataset. Figure 14 presents this result in purple, with the shaded area representing the standard error computed across five seeds. BAL-PM achieves on-par performance to this baseline, although it only requires 24000 data points (the full dataset contains 92,858 points). This result is another strong evidence of the sample efficiency of our method.

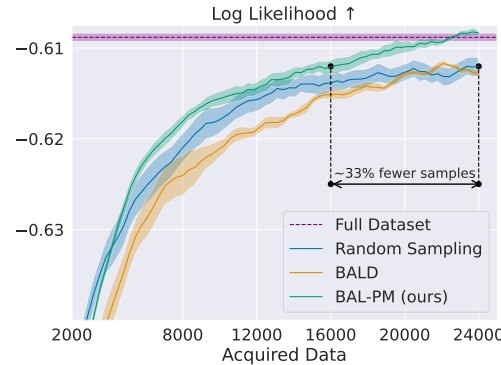

Figure 14: **Comparison with Preference Model trained on the full dataset.**

### J.2    Additional Data Selection Methods

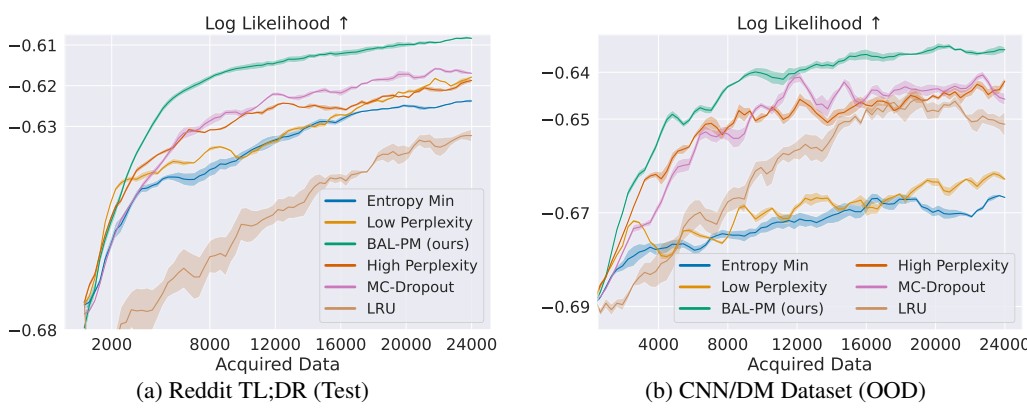

(a) Reddit TL;DR (Test)          (b) CNN/DM Dataset (OOD)

Figure 15: **Comparison with several additional baselines for Active Preference Modeling**. These baselines focus on different notions of uncertainty and diversity for acquiring samples.

We considered the following additional baselines:

- **Entropy Minimizer**: Entropy Minimizer: Inspired by Liu et al. [66], we consider an objective that, in addition to selecting points with high epistemic uncertainty, also selects points that are semantically similar to the current training points. This is equivalent to selecting points that increase the entropy of the prompt distribution **the least**, thus the name "Entropy Minimizer". It serves as a check for our central hypothesis that entropy maximization leads to better batch active learning.

- **Perplexity**: Inspired by Gonen et al. [67], we consider an objective that selects points based on the perplexity of the base LLM. We consider two versions: one that chooses points with lower perplexity (Low Perplexity) and another with higher perplexity (High Perplexity). This is an interesting baseline since perplexity is equivalent to the predictive entropy of the token distribution. Therefore, it helps to analyze how much the base LLM "knows what it does not know" in terms of preference modeling.

- **MC Dropout** [63]: This method performs approximate Bayesian inference via executing dropout at test time to generate different parameter hypotheses. Therefore, it can express epistemic uncertainty, which is used to select the most informative points.

- **Latent Reward Uncertainty** (LRU): This method computes a reward distribution over the data points by leveraging the latent reward model learned via the Bradley-Terry model.

Then, it selects extreme points (too high or too low rewards) as a proxy for the uncertainty of the model.

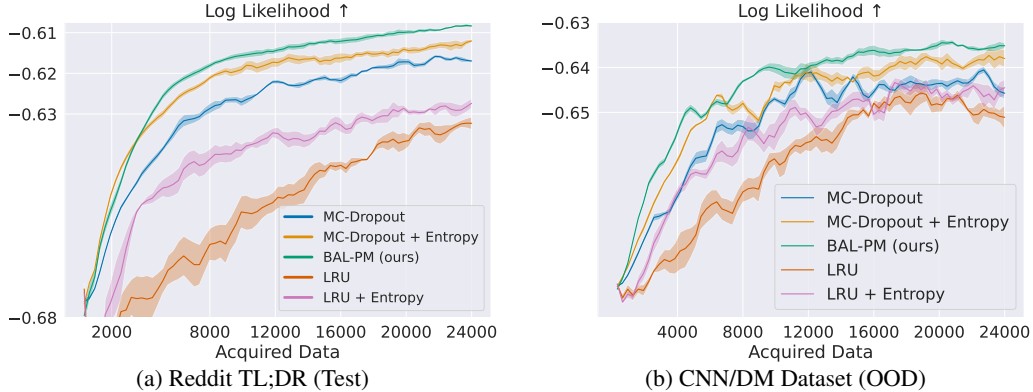

(a) Reddit TL;DR (Test)  (b) CNN/DM Dataset (OOD)

Figure 16: **The effect of incorporating the entropy objective in uncertainty baselines.** This shows how our proposed objective can also boost the performance of different baselines.

Figure 15 reports performances for both test and OOD sets. In both cases, BAL-PM outperforms additional baselines. In the sequence, MC-Dropout works best as the baseline that targets the epistemic uncertainty of a Bayesian model. Unsurprisingly, Entropy Minimizer and Low Perplexity work worse since they target points with lower entropy. LRU presented mixed results, suggesting that the latent reward may not represent well the preference model's uncertainty. More interestingly, while these models can represent different uncertainties to seek informative points, they naturally cannot provide in-batch diversity - they suffer from the same challenges as BALD. In this perspective, the BAL-PM objective can also improve upon those methods, as we show in Figure 16: we combined MC-Dropout and LRU with our entropy term to provide in-batch diversity, which consistently improved both methods across the datasets.

# K   Batch Samples

In this Section, we present some selected samples of the first acquired batch from BALD (Table 3) and BAL-PM (Table 4) for comparison. We sorted the points alphabetically to highlight duplicated prompts. BALD consistently acquires duplicated points, sometimes sampling more than ten times the same prompt. In contrast, BAL-PM samples semantically diverse points with no duplicates.

| BALD – Acquired Batch (Truncated Prompts) |
| --- |
| A bit of backstory: I've been in only 4 real long term relationships in my past.... |
| A bit of backstory: I've been in only 4 real long term relationships in my past.... |
| A bit of backstory: I've been in only 4 real long term relationships in my past.... |
| A few weeks ago my wife admitted to me that my best friend, (let's call him Marc... |
| A week ago I called off my relationship with my partner for a number of reasons,... |
| About a month ago my (23 F) boyfriend (26 M) of three and a half years and I got... |
| After 8 months my girlfriend decided to break up with me. Shes a very nice girl ... |
| For starters, its been awhile loseit, and I missed you! Things have been crazzzy... |
| For starters, its been awhile loseit, and I missed you! Things have been crazzzy... |
| For starters, its been awhile loseit, and I missed you! Things have been crazzzy... |
| For starters, its been awhile loseit, and I missed you! Things have been crazzzy... |
| Hello all I need some help regarding a friend of mine and a dream she had, well ... |
| Hello everyone, I am a student at a boarding school which means I am away from m... |
| Hi all. I am using a throwaway. I am 29f and my boyfriend is 32m. We have been d... |
| Hi all. I am using a throwaway. I am 29f and my boyfriend is 32m. We have been d... |
| Hi all. I am using a throwaway. I am 29f and my boyfriend is 32m. We have been d... |
| Hi first time user, and I am dyslexic so please forgive any spelling errors. T... |
| I am 31 years old and currently live in New York. I have been a professional tre... |
| I was sitting on a bus and the seat beside me was empty.. A young nun walked do... |
| I work inside of a bread depot, and the drivers are effectively brokers, or our ... |
| I work inside of a bread depot, and the drivers are effectively brokers, or our ... |
| I work inside of a bread depot, and the drivers are effectively brokers, or our ... |
| I work inside of a bread depot, and the drivers are effectively brokers, or our ... |
| I work inside of a bread depot, and the drivers are effectively brokers, or our ... |
| I work inside of a bread depot, and the drivers are effectively brokers, or our ... |
| I've been married to my husband for 3 years, it's been wonderful, I couldn't ask... |
| I've been married to my husband for 3 years, it's been wonderful, I couldn't ask... |
| I've been married to my husband for 3 years, it's been wonderful, I couldn't ask... |
| I've been married to my husband for 3 years, it's been wonderful, I couldn't ask... |
| I've been married to my husband for 3 years, it's been wonderful, I couldn't ask... |
| I've been married to my husband for 3 years, it's been wonderful, I couldn't ask... |
| It was my school's annual 5K, so the runners are students, faculty, and then ran... |
| Ive worked with this girl once a week for almost a year. When we met we were bot... |
| Ive worked with this girl once a week for almost a year. When we met we were bot... |
| Ive worked with this girl once a week for almost a year. When we met we were bot... |
| Ive worked with this girl once a week for almost a year. When we met we were bot... |
| Ive worked with this girl once a week for almost a year. When we met we were bot... |
| My girlfriend and I have been going out for about a year and have decided to mov... |
| My girlfriend and I have been going out for about a year and have decided to mov... |
| My girlfriend and I have been going out for about a year and have decided to mov... |
| My girlfriend and I have been going out for about a year and have decided to mov... |
| My girlfriend and I have been going out for about a year and have decided to mov... |
| My girlfriend and I have been going out for about a year and have decided to mov... |
| My girlfriend and I have been going out for about a year and have decided to mov... |
| My girlfriend and I have been going out for about a year and have decided to mov... |
| My girlfriend and I have been going out for about a year and have decided to mov... |
| My girlfriend and I have been going out for about a year and have decided to mov... |

Table 3: **First Acquired Batch by BALD (baseline)**.

## BAL-PM – Acquired Batch (Truncated Prompts)

\*\*Quick Background\*\*: As the title states, we've been together for 7 years datin...
\*\*The texts:\*\* Him: at least my mom thinks I'm cute me: I think you're cute ;)...
\*\*Warning: Avoid this film if you only broke up very recently! I advise this fil...
\*\*i(26m) have been dating her(26f) on and off for 5 years.\*\* I have come to the...
— So we broke up as in words she had severe depression and it wasn't fair to m...
A little while back, my sister asked me why some men were homophobic. I answere...
A small background. I live in in Puerto Rico, where I haven't had to good an ex...
About a month ago we started having problems with our cable. The picture would g...
Background info: Little background. I started medical school a few years back. I...
Backstory: I'm a 17 year old student in the U.K. currently in sixth-form. Back i...
Be sure to explain in detail with line breaks. Hey my name is Matt and i honest...
Because I live in a very conservative Catholic neighborhood, I cannot come out a...
Hello all, Story: I played around with some stocks a few years back buying ...
Hello reddit My LDR girlfriend of six months told me yesterday that she wasn't ...
Hello! Last group of friends I had was back in 10th Grade. Since then my depre...
Hello! I'm a 23 y/o F dating a 30 y/o male. This is by far the best relationship...
Hello, I'm kind of new to this sub reddit but I figured I'd get an opinion from ...
Hey Reddit. I spent at least 20 mins looking for the correct sub-reddit for men'...
Hey all. My classmates and I at the SUNY Purchase Film Conservatory are in the p...
Hey everyone so I'm about 3 months in of my 6 month regimen before I get gastric...
Hey fellow revenge-lovers, here's a quick one, that happened about an hour ago. ...
Hey guys this is strange to begin with, but I"ll introduce the situation: I'm ...
I am a 24 y/o male and I have been dating a girl who is 22 years old for about 1...
I am an 18 year old college student and I have no attachments to my local area. ...
I am aware that this has been proposed before. I personally believe that it woul...
I am dating a complete dime like I get compliments all the time about her from s...
I can't focus. I can't become and remain motivated. When I've learned something ...
I know we are young but bear with me, I didn't know where else to go for this ty...
I live in California and am the co-owner of a car, with the names on the title b...
I made a previous post here but it sounded kind of stupid with the way I phrased...
I met my current girlfriend in highschool. She's the only woman I've ever been ...
I should start with saying neither of us have had a chance to travel anywhere ex...
I want to start off by saying I love my SO and I'm looking for suggestions befor...
I was in a pretty serious car accident this week, and my car was easily totaled....
I will try to make this as short as possible. a long time ago i met this girl, ...
I'll keep it short :3 I'm 18, he's 18. Dating for 3 years. When we walk togethe...
I'll keep this as succinct as possible. I moved in Sept. 1. I used to live here...
I've been with my boyfriend for 4 years, it hasn't been the best relationship, b...
I've been with my gf for almost 7 years. Lived together for about 5 years. A few...
I've been working with this girl for 2 months. it started at work where i was he...
If you want to understand the scam, here's what's happening: Okay, so I found a...
Im 27. Single. I am a productive member of society. I work full time i pay my ow...
It was New Year's Eve and my family was driving off to my grandparents' house. H...
It wasn't that long term relationships but we lived together for 6 months so we ...
Just got the new Kobo touch and they provided me with a $10 gift card for their ...
My friend's little brother is really suffering in his dorm. He's lost 15-20 poun...
My girlfriend an I have been dating for three years. Its been the best time of m...
My girlfriend and I met through family friends a year and a half ago. We've been...

Table 4: **First Acquired Batch by BAL-PM**.

