# OpenReview forum: "Deep Bayesian Active Learning for Preference Modeling in Large Language Models"
_NeurIPS.cc/2024/Conference — NeurIPS 2024 poster_

### Official Review · Reviewer_tUWV · 2024-07-10

**Soundness:** 3
**Presentation:** 3
**Contribution:** 2
**Rating:** 5
**Confidence:** 3

**Summary:**

This work proposes BAL-PM, a stochastic acquisition policy aiming to select informative data samples (i.e., the prompt and its corresponding paired responses) that need to collect human feedback for LLMs' preference learning.
Specifically, BAL-PM mainly addresses the issue of sampling redundant data points in previous methods that rely solely on epistemic uncertainty, such as BALD. On top of the previous BALD acquisition method, BAL-PM employs a task-agnostic entropy estimator for the prompt distribution to enhance the diversity of acquired data samples.
Experiments on the Reddit and CNN/DM datasets under a pool-based setting demonstrate the effectiveness of BAL-PM over random sampling and the BALD baseline.

**Strengths:**

- The paper is easy to follow.
- The method is well-motivated by the observation of the tendency of sampling redundant data points in previous methods.
- The effectiveness of key components of the proposed BAL-PM is well justified and supported by the experimental results and ablation.

**Weaknesses:**

- The experiment part is not compelling and clear enough to me. Regarding the reported log-likelihood metric, it is unclear to me how it is calculated (e.g., on preferred response?) and why it is the only metric used for evaluating preference modeling. It seems to me that it can not measure to what extent the chosen response is preferred over the rejected one. Regarding the datasets evaluated, the task is limited to text summarization. The experiment part can be improved by adopting more metrics on more general-purposed datasets/tasks to illustrate the effectiveness of the proposed method.
- While encouraging data diversity is plausible and effective in LLM's preference learning according to the experiments, the necessity of using the feature of the base LLM to measure the entropy of the prompt distribution is unclear to me. Although it is mentioned as a common approach in lines 97-99, the motivation for focusing on feature space is still somewhat vague to me and may introduce extremely high computational overhead in the context of LLM's preference learning.
- The proposed BAL-PM, which requires maintaining/updating an ensemble of adapters for collecting each batch, may be difficult to practice in large scale.
- The discussion of prior works is inadequate. Besides the difference in configuration, the technical parts of other related active preference optimization of LLMs should be discussed more.

**Questions:**

- What is the advantage of adopting the (static) feature space of the base LLM for estimating the entropy of the prompt distribution? Is there any evidence to demonstrate that it is superior to other approaches, such as using smaller proxy models (e.g., BERTScore) or simpler fashions?
- I am curious about whether different adapters can make diverse predictive distributions with limited expressive power. Is there any empirical evidence for this?

**Limitations:**

The authors have adequately addressed the limitations and potential negative societal impact of their work.

---

> ### Author Rebuttal · Authors · 2024-08-07
>
> Thank you for sharing your concerns and questions. We address them below:
>
> **Q14** Why is the Log-Likelihood metric used for evaluating Preference Modeling?
> > We refer to **Q1** (global rebuttal).
>
> **W3** The task is limited to text summarization.
> > **R3**  We understand the importance of different NLP tasks to enrich evaluation. Still, we focus on validating the BAL-PM as an Active Learner in Preference Modeling. We argue that the considered datasets already bring the key challenges for this task to provide empirical evidence to support our findings:
> > - They present real human feedback, reflecting the true human behavior and aleatoric uncertainty involved in the preference elicitation process;
> > - Prompts are human-generated and go beyond simple instructions. They incorporate human subjectivity and diversity in terms of culture and topics, requiring a high level of language understanding. We refer to Tables 25-28 in [1] for samples. It is a challenging setup.
> > - The CNN/DM News dataset presents OOD prompts (from News, far from Reddit posts) while following the same label-generation process. It allows the evaluation of robustness/generalization of learned preference models.
>
> **W4** The experiment part can be improved by adopting more metrics on more general-purposed datasets/tasks to illustrate the effectiveness of the method.
> > **R4** We highlight that our work already explores several dimensions to validate BAL-PM, going beyond performance claims. For instance, we:
> > - Provide evidence in an OOD setting (Figs 4b, 5b) to validate robustness/generalization;
> > - Present measures of diversity to analyze the acquisition of redundant samples (Fig 6);
> > - Evaluate the scalability of our method for larger LMs with 70b+ parameters (Fig 7);
> > - Evaluate crucial design choices related to the policy objective and its implementation (Figs 8-10);
> > - Analyze how the stochastic policy balances the influence of each epistemic uncertainty source, a key property of BAL-PM (Fig 11).
> >
> > All raised experiments validate crucial points of BAL-PM and Active Preference Modeling, providing evidence to support our method.
>
> **Q15** What is the advantage of adopting the feature space of the LLM for entropy estimation of the prompt distribution? Is it superior to using smaller proxy models (e.g., BERTScore) or simpler fashions?
> > **A15** Estimating the entropy of the prompts requires representations that encode semantic features. The underlying idea follows the core of BERTScore: leveraging the distances on this semantic vector space to represent similarities among natural language sentences. We expand this perspective to use not as a metric but as a density/entropy estimator of the prompt distribution.
> > The key point is that **this process requires a feature space that can perform well in encoding the extent of the natural language distribution**. Simpler feature extractors, such as tf-idf or CBoW, struggle to encode contextual semantic information. While traditional deep learning models like BERT and ELMo have promising architectures and learning objectives, they are currently limited by their model and training set sizes compared to LLMs trained on meticulously curated datasets that encompass the entire web. This inherent relative limitation translates to performance: for evidence, we refer to the SuperGLUE benchmark [2], comprising several language understanding tasks and online leaderboard. Currently, the best ranked models present 10b+ parameters (the scale of our experiments). BERT-based baselines and classic techniques place in 26th position (see “SuperGLUE baselines”). This result strongly suggests that LLMs provide better representations than models with smaller scales, which is especially crucial for our method, particularly for the challenging prompts in the considered datasets.
>
> **W5** Focus on the LLM feature space to measure entropy may introduce computational overhead in the context of LLM's preference learning.
> > **R5** We refer to **Q2** (global response) for computational cost clarifications. There is no additional cost as we can extract these "entropy features" and "preference model features" simultaneously.
>
> **W6** BAL-PM requires maintaining/updating an ensemble of adapters for collecting each batch, which may be difficult to practice in large scale.
> > **R6** We again refer to **Q2**. Our adapters are simple MLPs with 2 hidden layers, whose cost is reasonably cheap for LLM research.
>
> **W7** The discussion of prior works is inadequate and should describe the technical parts of Active Preference Optimization in LLMs.
> > **R7** We extended the "Active Preference Modeling" section in Related Work, detailing the draft references [11, 13, 25, 35]. The actual text is too long to fit here, so we summarize: [11, 13, 25] theoretically studies active query generation and propose different methods based on estimating confidence bands to generate high-uncertainty triples. [35] generate answers using double Thompson Sampling, representing model uncertainty with an Epistemic Neural Network. We also describe that [18] uses a fully fine-tuned deep ensemble for model uncertainty estimation. Let us know if this is clear. Otherwise, we may paste the exact text as a comment.
>
>
> **Q16** Can adapters make diverse predictive distributions with limited expressive power?
> > **A16** Adapters can be seen as a special case of Epistemic Neural Networks [3], where the prior distribution is a mixture of delta functions over indices. EpiNets provide principled theoretical justification for adapters. There is empirical evidence of adapters in the context of exploration in LLMs [4], which is similar to our method (on capturing epistemic uncertainty) but focused on answer generation, not data selection. We also incorporated MC-Dropout [5] as baseline (see **Q6**), a well-known method for Bayesian approximation. Adapters outperform MC-Dropout in our setup, suggesting effectiveness on approximating posterior distributions.

---

> > ### Comment · Reviewer_tUWV · 2024-08-11
> > **Reply to the Rebuttal**
> >
> > The authors did a good job of addressing most of my concerns and making clarifications regarding their method and evaluation. However, I strongly encourage the authors to validate the effectiveness of BAL-PM on a broader range of tasks/datasets, especially given the notable computational efficiency of the proposed method and the moderate scale of common academic preference datasets. In my humble opinion, the current paper is a nice technical contribution with fair empirical significance to the community.

---

> ### Author Response · Authors · 2024-08-07
> **Rebuttal References**
>
> [1] Stiennon et. al. Learning to summarize with human feedback. NeurIPS, 2021.
>
> [2] Wang et. al. SuperGLUE: A Stickier Benchmark for General-Purpose Language Understanding Systems. NeurIPS, 2019.
>
> [3] Osband et. al. Epistemic Neural Networks. NeurIPS, 2023.
>
> [4] Dwaracherla et. al. Efficient Exploration for LLMs. ICML, 2024.
>
> [5] Gal et. al. Dropout as a Bayesian Approximation:
> Representing Model Uncertainty in Deep Learning. ICML, 2016.

---

### Official Review · Reviewer_vibJ · 2024-07-13

**Soundness:** 4
**Presentation:** 4
**Contribution:** 3
**Rating:** 6
**Confidence:** 4

**Summary:**

&nbsp;

The authors present a method, Bayesian Active Learning for Preference Modeling (BAL-PM) that seeks to learn a preference model in a sample-efficient manner within an active learning setting. A key insight by the authors is the consideration of task-dependent and task-agnostic uncertainty to encourage diversity in the sampled prompt-completion pairs. My main concern with the approach is the link between the pool-based active learning setting and active learning over an open-ended and continuous prompt space which I will describe below. As such, my recommendation is borderline pending the authors rebuttal on this point.

&nbsp;

**Strengths:**

&nbsp;

The paper is well-written and the idea behind BAL-PM is well-motivated. The empirical results appear compelling in the pool-based setting and furthermore, the authors provide the code to reproduce their results. The analysis of the diversity of the acquired prompts is a useful diagnostic to validate the authors' hypothesis that BAL-PM attains performance improvements by encouraging diversity in the sampled prompts.

&nbsp;

**Weaknesses:**

&nbsp;

__MAJOR POINTS__

&nbsp;

1. My main concern with the current work is the link to the real-world problem of constructing a  preference model in a sample-efficient fashion in the absence of a pool of labelled prompt-completion pairs. Specifically, as I understand, one of the main motivations for being sample efficient in constructing the preference model is the cost of human labelling. In the experiments considered by the authors, the datasets already contain labels. As such, I would ask  a) what is the motivation for being sample-efficient in the pool-based active learning setting? b) assuming the authors feel performance on the pool based setting is representative of an open-ended active learning problem, how would they go about verifying that BAL-PM performs well in the non pool-based setting?

&nbsp;

__MINOR POINTS__

&nbsp;

1. It would be great if the references appeared in numbered order.

2. There are some missing capitalizations in the references e.g. "Bayesian".

3. The arXiv identifier is missing for reference 2.

4. When referencing PyTorch, [1] should be used rather than the earlier workshop paper version.

5. It would be great if the authors could provide instructions for reproducing the experimental results in the README of their GitHub repository.

6. Reference 7 was accepted at TMLR.

7. Line 38, I would not expect the statistic that agreement rates among human labelers is typically 60-75% to be robust across all problem settings. It would be worth qualifying this statement.

8. The arXiv identifier for reference 18 is missing.

9. "Prompt-completion pairs" would potentially be a more appropriate terminology in place of "prompt-answer" pairs.

10. On line 105, S is not defined, this should presumably be \Chi.

11. Line 134 typo, "an unsupervised".

12. In Figure 4, it would be great to include in the caption the number of random trials for which the errorbars are computed.

13. In the abstract the authors do not mention that the figures of 33% and 68% apply to a random sampling baseline.

&nbsp;

__REFERENCES__

&nbsp;

[1] Paszke, A., Gross, S., Massa, F., Lerer, A., Bradbury, J., Chanan, G., Killeen, T., Lin, Z., Gimelshein, N., Antiga, L. and Desmaison, A., 2019. PyTorch: An imperative style, high-performance deep learning library. Advances in Neural Information Processing Systems, 32.

[2] Yang, A.X., Robeyns, M., Coste, T., Wang, J., Bou-Ammar, H. and Aitchison, L., 2024. Bayesian reward models for LLM alignment. arXiv preprint arXiv:2402.13210.

&nbsp;

**Questions:**

&nbsp;

1. In Section 4, the authors mention that their Bayesian preference model is constructed using an ensemble of adapters. What was the motivation for this choice over Laplace-LoRA [2] for example?

2. In Figure 7, random sampling outperforms BAL-PM for the 140b parameter model early in the active learning trace. Do the authors have an explanation for why this might be the case?

3. It would be interesting to see the full details of the validation procedure for choosing the hyperparameters, most notably the entropy term \Beta. How sensitive was performance to the choice of \Beta hyperparameter? What is the link to Section F of the appendix?

4. In terms of the results presented in Figure 9, how do the authors compute the aleatoric uncertainty for the datasets? Are these uncertainty estimates provided in the datasets?


&nbsp;

**Limitations:**

&nbsp;

The biggest limitation I foresee with the current work is the potential for the performance of BAL-PM to carry over between the pool-based active learning setting and the non pool-based setting as described above.

&nbsp;

---

> ### Author Rebuttal · Authors · 2024-08-07
>
> Thank you for sharing your concerns and questions. We address them below:
>
> **Q8** What is the motivation for being sample-efficient in the pool-based active learning setting?
>
> > **A8** We clarify that the goal of a pool-based setup is to mimic an open-ended data selection setup. Although we use logged data in the pool, **the experiment only reveals the labels when the active learning policy selects the data points**. Hence, for the Preference Model, the pool data is completely unlabeled. It is a standard experimentation protocol in Active Learning literature [1,2] and allows the use of real human feedback. For instance, Gleave [3] also adopts the same experimental setup.
> > A non-pool-based setup would require either collecting human feedback for every batch in each experiment (impractical) or relying on a preference simulator that tries to mimic human behavior. This is also unrealistic, as accurately modeling the human behavior and the aleatoric uncertainty involved in the preference elicitation process is really challenging. Therefore, using data that leverages real human preferences is more realistic for Preference Modeling and enables purely offline experiments.
>
>
> **Q9** How would BAL-PM perform in a non-pool-based setup?
> > **A9**  Given answer **A8**, the key thing here is to realize that the difference between a pool-based and a non-pool-based one is that the latter permits generating different answers for the prompts, while in the former the answers are fixed. As pointed out by previous work [3], this means that our setup may actually **underestimate** the benefit of Active Learners (BAL-PM) since we limit the stochastic policy to prompt selection. Therefore, in a non-pool setup, we believe that BAL-PM should work as well as, or even better, given another degree of freedom for data selection. Crucially, we designed BAL-PM to work in both settings without changes.
>
>
> **W3** Minor Points
> > **R3** We appreciate the detailed list. We added all points to our current draft, which should be reflected in the camera-ready version. For point 7, we qualified the statement by clarifying that this is the result observed in past preference modeling settings for LLM finetuning (as in the references).
>
> **Q10** What is the motivation of using adapters over other methods such Laplace-LoRA?
> > **A10** This is a great question that lies at the core of our method. The choice of adapters is computational tractability: they rely on LLMs solely for feature extraction. Hence, BAL-PM does not require training or finetuning LLMs during the active learning loops, considerably reducing the computational costs and allowing scaling for very large LMs. A method like Laplace-LoRA (or any LoRA method) requires finetuning the base LLM. Even if we assume that the finetuning is cheap (which is not for 70B+ models), updating the LLM requires generating new features for all the prompt-answer pairs in the pool at each training loop. This considerably increases the computational costs of performing active learning loops. For adapters, we can generate features once *before* the active learning experiment in inference-optimized settings (with quantization, for instance), and go beyond 100B models in a single A100 GPU.
>
> **Q11**  In Fig 7, random sampling outperforms BAL-PM for the 140b parameter model early in the active learning trace. Do you have an explanation for why this might be the case?
> > **A11** We hypothesize that this may be due to 4-bit quantization affecting prompt representations for entropy estimates, which is crucial in the early stages of training for accelerating the Bayesian Preference Model learning. The quality of the features is one of the limitations of BAL-PM. Also, as the Bayesian Preference Model contains more parameters for larger base LMs, it may require more initial data to fit well and provide accurate epistemic uncertainty estimates.
>
> **Q12** What is the validation procedure for hyperparameter selection? How sensitive BAL-PM is for the choice of $\beta$? What is the link with the Appendix F?
> > **A12** The procedure was quite standard: given the search space in Table 2, we evaluated the candidates in the held-out validation set and selected the best-performing hyperparameter. We tuned the presented hyperparameters in isolation (a grid search would be too expensive).
> > **We perform a $\beta$ robustness analysis in Fig. 15 (Rebuttal PDF)**, considering the values in the search space. The impact of the choice is more noticeable to values 100x greater/lower than the optimal choice. Values around 10x greater/lower still perform well, suggesting good room for choosing this hyperparameter. Furthermore, we employed the same value of $\beta$ across the different datasets and LLMs, suggesting robustness across different relevant dimensions.
> > Crucially, $\beta$ trades off the contribution of two different terms. As such, it provides a spectrum of objectives and may recover the two extremes presented in the ablation of Fig. 8. Naturally, different choices of $\beta$ will change the uncertainty score ratio presented in Fig. 11 on Appendix F (i.e., the contribution of each term after convergence). Nevertheless, and most importantly, the behavior of the curves – the entropy contribution progressively reducing and converging and the relevance of epistemic uncertainty estimates increasing – should remain.
>
> **Q13** How do you compute the aleatoric uncertainty? Is this provided in the datasets?
> > **A13** We compute the predictive uncertainty as the entropy of the posterior predictive distribution (the first entropy term in the Equation 3). Similarly, we compute the epistemic uncertainty via Equation 3. By the Law of Total Variance, the Total (Predictive) Uncertainty in a Bayesian Model is the sum of the Epistemic Uncertainty and Aleatoric Uncertainty. Thus, we can estimate the Aleatoric Uncertainty based on that. These are only **estimates** from our model. Thus, they are not values provided in the dataset.

---

> > ### Comment · Reviewer_vibJ · 2024-08-12
> > **Many Thanks to the Authors for their Rebuttal**
> >
> > &nbsp;
> >
> > Many thanks to the authors for their rebuttal.
> >
> > 1. In terms of the pool-based vs. non-pool-based setting, I think it would be beneficial to include a preference simulator for the non-pool-based setting which would a) emulate an open-ended active learning loop b) allow some simulated notion of ground truth aleatoric uncertainty to be present. I would encourage the authors to consider this for the camera-ready version of the paper.
> >
> > 2. Many thanks for the clarification on the use of adapters in place of Bayesian LoRA. The intuition regarding finetuning the base LLM makes a lot of sense and I can see why this would be an important factor to stabilize the surrogate in an active learning loop.
> >
> > 3. The intuition for the performance of BAL-PM relative to random sampling early in the trace makes a lot of sense.
> >
> > 4. Many thanks for the sensitivity analysis on the \Beta parameter.
> >
> > Given the above, I am happy to raise my score but I would strongly encourage the authors to think about including an additional experiment emulating an open-ended active learning loop with a preference simulator. I think this would "complete" the paper in the sense that both a) the pool-based setting with realistic human preferences and b) the non-pool-based setting with synthetic preferences are considered.
> >
> > &nbsp;

---

> ### Author Response · Authors · 2024-08-07
> **References**
>
> [1] Smith et. al. Prediction-Oriented Bayesian Active Learning. AISTATS, 2023.
>
> [2] Imberg et. al. Optimal sampling in unbiased active learning. AISTATS, 2020.
>
> [3] Gleave et. al. Uncertainty Estimation for Language Reward Models, 2022.

---

### Official Review · Reviewer_aVyo · 2024-07-13

**Soundness:** 3
**Presentation:** 3
**Contribution:** 3
**Rating:** 6
**Confidence:** 4

**Summary:**

This paper proposed a novel framework to select the most informative preference data for training based on Bayesian active learning.
To collect the prompt-response pair (x,y), it firstly selects the prompt based on Bayesian Active Learning by Disagreement by maxmizing the information gain. Then, the selection of response considers both the preference model epistemic uncertainty estimation and entropy estimation for the acquired prompt distribution, which addresses the challenges of selecting diverse samples. When evaluated on Reddit and CNN/DM preference datasets, it saves 33% and 68% training samples by comparing to random sampling.

**Strengths:**

1. The research problem is important to data-efficient training.
2.  The presented emprical results are impressive in saving utilised training data over listed baselines.
3.  The empirical studies is comprehensive and verify the effectiveness across different settings.

**Weaknesses:**

1. Lack for diversity-based and uncertainty-based sample selection methods, such as [1,2,3,4], which makes the technical contributions less reliable,
2. As the method is based on feature extraction from existing LLMs, the computation cost in estimating the diversity and uncertainty is not clear, which inhits its pratical application.


References:
[1] What Makes Good In-Context Examples for GPT-3?
[2] An Information-theoretic Approach to Prompt Engineering Without Ground Truth Labels
[3] Demystifying Prompts in Language Models via Perplexity Estimation
[4] Diverse Demonstrations Improve In-context Compositional Generalization

**Questions:**

See the details in Weakness.
1. How are the performances of existing other baselines.
2. What are the computation cost in estimate the utility score, diversity and entropy.

**Limitations:**

Yes, the authors mentioned the intrinsic limitation of the transformer's embedding, i.e., noise-TV. The impropriate selected samples can be toxic, unfair or not well-represented.

---

> ### Author Rebuttal · Authors · 2024-08-07
>
> Thank you for highlighting the strengths of our work and for bringing up your concerns and questions. We aim to address them in this response.
>
> **Q6** How does BAL-PM perform in comparison with other sampling methods [1, 2, 3, 4]:
>
> > **A6** Thank you for the references. We highlight that these works focus on data sampling for In-Context Learning (ICL), while ours investigate Active Learning for Preference Modeling in LLMs. While both topics aim to improve sample efficiency, they operate in different setups: **ICL is a test-time procedure** that incorporates training points to condition the prompt in order to improve predictions of a test point), while **Active Learning is a training-time procedure** that selects training points for improving the model. Fundamentally, this may lead to different conclusions as to what is required for sample efficiency. Nonetheless, we adapted most of the suggested methods to our scenario to evaluate them and provide more evidence of our contribution. We also added other baselines from the Active Learning literature:
> >  - Entropy Minimizer: Inspired by [1], we consider an objective that, in addition to selecting points with high epistemic uncertainty, also selects points that are semantically similar to the current training points. This is equivalent to selecting points that increase the entropy of the prompt distribution **the least**, thus the name "Entropy Minimizer". It serves as a check for our central hypothesis that entropy maximization leads to better batch active learning.
> >  - Perplexity: Inspired by [3], we consider an objective that selects points based on the perplexity of the base LLM. We consider two versions: one that chooses points with lower perplexity (Low Perplexity), and another with higher perplexity (High Perplexity). This is an interesting baseline since perplexity is equivalent to the predictive entropy of the token distribution. Therefore, it helps to analyze how much the base LLM "knows what it does not know" in terms of preference modeling.
> >  - MC Dropout [5]: This method performs approximate Bayesian inference via executing dropout at test time to generate different parameter hypotheses. Therefore, it can express epistemic uncertainty, which is used to select the most informative points.
> >  - Latent Reward Uncertainty (LRU): This method computes a reward distribution over the data points by leveraging the latent reward model learned via the Bradley-Terry model. Then, it selects extreme points (too high or too low rewards) as a proxy for the uncertainty of the model.
> >
> > Figure 16 (Rebuttal PDF) reports performances for both test and OOD sets. In both cases, BAL-PM outperforms additional baselines. In the sequence, MC-Dropout works best, as the baseline that targets the epistemic uncertainty of a Bayesian model. Expectedly, Entropy Minimizer and Low Perplexity work worse since they target points with lower entropy. LRU presented mixed results, suggesting that the latent reward may not represent well the preference model's uncertainty. More interestingly, while these models can represent different uncertainties to seek informative points, they naturally cannot provide in-batch diversity - they suffer from the same challenges as BALD. In this perspective, the BAL-PM objective can also improve upon those methods, as we show in Figure 17: we combined MC-Dropout and LRU with our entropy term to provide in-batch diversity, which consistently improved both methods across the datasets.
> > Lastly, we highlight that the suggested method in [2] is akin to BALD, as it focuses on the mutual information objective (described in Equation 3 in our paper). [4] focus on semantic parsing approaches to enable compositional generalization, which implicitly assumes key properties of the considered input: For instance, there is a mapping between the natural language sentences and formal queries. In preference modeling for LLMs, we do not assume the existence of such formal queries. Given this assumption, [4] builds diversity sampling on top of syntax trees, which is out of the scope of our work on preference modeling.
>
>
> **Q7** What are the computation costs involved in BAL-PM?
> > **A7** We kindly refer the reviewer to **Q2** in our global response, where we discuss in detail the costs of our method, contextualizing with the Active Learning domain.
>
> **W2** Limitation: the impropriate selected samples can be toxic, unfair or not well-represented
> > **R2** We provide some context around this potential limitation. In terms of toxicity, we highlight that selecting toxic prompts for getting preferences is actually a crucial step for detoxifying fine-tuned language models. If the reward model is not trained on such prompts, they may behave as out-of-distribution samples, leading to incorrect predictions that could reinforce toxic answers. Therefore, providing preference labels that consider safe answers over toxic ones is necessary for preference modeling. A good example is the work of Bai et. al. [5], where the employed dataset contains content that may be offensive or upsetting, to make LMs less harmful.
> > Regarding representativity, we believe that having an entropy term that seeks diversity in the prompts will actually reinforce representativity over different prompt classes.
>
> We hope that we addressed your questions and concerns. Let us know if there are any further points to be clarified - we genuinely appreciate your review.
>
> [1] Liu et. al. What Makes Good In-Context Examples for GPT-3?. ACL DeeLIO, 2022
>
> [2] Sorensen et. al. An Information-theoretic Approach to Prompt Engineering Without Ground Truth Labels. ACL, 2022
>
> [3] Gonen et. al. Demystifying Prompts in Language Models via Perplexity Estimation. EMNLP Findings, 2023
>
> [4] Levy et. al. Diverse Demonstrations Improve In-context Compositional Generalization. ACL, 2023.
>
> [5] Bai et.al. (Anthropic Team). Training a Helpful and Harmless Assistant with Reinforcement Learning from Human Feedback, 2022.

---

> > ### Comment · Reviewer_aVyo · 2024-08-10
> >
> > Thanks for the detailed responses, which largely address my concern. I will raise the score accordingly.

---

### Official Review · Reviewer_x99G · 2024-07-17

**Soundness:** 4
**Presentation:** 4
**Contribution:** 3
**Rating:** 7
**Confidence:** 3

**Summary:**

This paper presents BAL-PM, a Bayesian active learning framework for the training preference model. The authors propose an acquisition policy that seeks examples that have high epistemic uncertainty and can maximize training data’s entropy. Specifically, the epistemic uncertainty is estimated by the training preference model and entropy is estimated by the base LLM. Experiments on two preference datasets show that the proposed method requires fewer training data compared to previous methods (33% and 68% reduction on each dataset accordingly). The author also shows that their method is scalable to larger LLMs.

**Strengths:**

- The problem is clearly formulated. The method is well derived and justified.
- The results are strong, showing significant improvement in sample efficiency.
- The authors test their method on models of various sizes, demonstrating its scalability.
- The paper is clearly written and easy to follow.

**Weaknesses:**

- The evaluation metric is limited. The authors only show the log-likelihood of the test sets. This single metric has no guarantee that preference models trained by this method have a more accurate judgment (shown by pair preference accuracy) and lead to better fine-tuned LMs (shown by tuning with the trained preference model)

**Questions:**

- How do the results compare to models trained on the whole dataset?
- Do the trained models lead to better LMs?

**Limitations:**

The authors have properly adequately the limitations of their work in the paper.

---

> ### Author Rebuttal · Authors · 2024-08-07
>
> Thank you for appreciating our work strengths and for raising concerns and questions. We hope to clarify them in this response. Please see the answers below:
>
> **W1** "The evaluation metric is limited and gives no guarantee of more accurate judgment or better fine-tuned LMs."
> > **R1** We kindly refer the reviewer to **Q1** under the global rebuttal, where we clarify the Log Likelihood metric and shows that it provides more information about preference strength for ranking models than prediction accuracy. We also refer to **Q5** where we argue about the relationship between Average Log-Likelihood as a performance measure for preference models and the quality of the fine-tuned policy.
>
> **Q4** How do the results compare to models trained on the whole dataset?
> > **A4** In Figure 12 of the Rebuttal PDF, we show (in purple) the performance of a trained model in the full dataset (over five seeds). BAL-PM achieves on-par performance only requiring ~24000 points (the full dataset contains 92,858 points). This result is another interesting evidence of the sample efficiency of our method.
>
> **Q5** Do the models evaluated by Average Log Likelihood (LL) lead to better finetuned policies?
>
> > **A5** Despite our work strictly focusing on Preference Modeling (which has several applications in psychology [1], economy [2], and sociology [3] that goes beyond RLHF), we agree that fine-tuning LM policies is a very relevant downstream task. Our Preference Modeling optimization objective and model selection protocol follow exactly the prior influential work on the topic [4, 5], which provides evidence that better preference models (in terms of validation loss) lead to improved downstream policies. Thus, we expect our models to behave similarly under the same conditions.
> >
> > As additional evidence, we empirically illustrate the relationship between log-likelihood and policy performance on a simplified setup (see Figure 13 on Rebuttal PDF). Here, prompts $x$ and answers $y$ are real numbers in [0, 1]. The ground-truth reward function is given by a Gaussian density function $r(x, y) = \mathcal{N}(x + y \mid \mu = 1.0, \sigma = 0.4)$, and true preferences follow the Bradley-Terry model. In this setup, we progressively increase the training set size (the x-axis in Figure 13a) in which we train the preference models. This process generates different models with increasing levels of test-set average log-likelihood. Then, similar to [6], we optimize the base policy via a Best-of-N optimizer by leveraging each of these learned preference models. Finally, we report the rate in which the fine-tuned policy answer is preferable over the base policy answer according to the ground truth reward model ("win rate"). Although simple, this setting allows us to bypass several optimization and distributional challenges and solely focus on evaluating the relationship between average log-likelihood and the performance of the fine-tuned policy. Figure 13 (left) reports the log-likelihood (red) and the win rate against the base policy (blue). Figure 13 (right) directly plots both measures and fits a regression line. We observe a strong correlation, which aligns with our point: **a higher test-set average log-likelihood means that the preference model is better at predicting the ground truth preferences, assigning higher rewards for better answers, and, therefore, improving fine-tuned policies that maximize such reward scores.**
>
>
> We hope we clarified your questions and concerns. Please let us know if there any any further points to be clarified - we genuinely appreciate your time reviewing our work.
>
> **References**
>
> [1] Kahneman et. al. Judgement under Uncertainty — Heuristics and Biases, 1981.
>
> [2] Armstrong, W.E. A note on the theory of consumer’s behavior. Oxford Economics, 1950.
>
> [3] Sen, A.K. Social choice theory. Handbook of Mathematical Economics, 1986.
>
> [4] Ziegler et. al. Fine-tuning language models from human preferences, 2019.
>
> [5] Stiennon et. al. Learning to summarize with human feedback. NeurIPS, 2021.
>
> [6] Gao et. al. Scaling laws for reward model overoptimization. ICML, 2023.

---

> > ### Comment · Reviewer_x99G · 2024-08-12
> >
> > Thank you for your response and the additional results. My concerns are addressed. I'll maintain my overall rating as I've already given an acceptance score. But I will increase my Soundness rating.

---

### Author Rebuttal · Authors · 2024-08-07

We thank the reviewers for raising concerns and providing feedback to improve our work. We appreciate the acknowledgement that:
- **The paper is clear and well-written** (x99G, vibJ, tUWV);
- **The proposed method is principled and well-motivated** (all reviewers);
- **Empirical results are strong/impressive/compelling/effective** (all reviewers);
- **The method demonstrated scalability (x99G) and effectiveness across different settings** (aVyo).

We highlight the new empirical evidence in the Rebuttal PDF to clarify concerns and incorporate feedback. In detail:

- **The performance of preference models in the full dataset** (Fig. 12, **Q4**, x99G);
- **The relationship between a preference model test loglikelihood and the corresponding finetuned policy performance** (Fig. 13, **Q5**, x99G)
- **The objective's $\beta$ hyperparameter robustness analysis** (Fig. 14, **Q12**, vibJ)
- **Several additional baselines** (Figs. 15/16, **Q6**, aVyo)

We now clarify questions raised by different reviewers:

**Q1** Is Average Log Likelihood (LL) a proper performance measure for Active Preference Modeling? **(Reviewers x99G, tUWV)**

 > **A1** We first clarify how the Average LL is computed. Given the test set $\mathcal{D}\_{test} = \\{(x, y_1, y_2, {y_1 \succ y_2})\\}^{N}$ and the learned preference model $p_{\boldsymbol{\theta}}(y_{1} \succ y_{2} \mid x, y_{1}, y_{2})$, the average Log Likelihood is given by $LL(\mathcal{D}\_{test}, \boldsymbol{\theta}) = \mathbb{E}\_{(x, y_1, y_2, y_1 \succ y_2) \sim \mathcal{D}\_{test}} [\log(p_{\boldsymbol{\theta}}(y_{1} \succ y_{2} \mid x, y_{1}, y_{2}))]$. It is exactly the objective maximized in standard binary classification (or, equally, the minimization of the negative log-likelihood loss) but computed over the test data. In other words, this is the negative "test loss".
 >
 >  Average LL is a typical metric in the Active Learning and Uncertainty Quantification literature [1, 2, 3]. For Preference Modeling, it is very relevant as **LL directly accounts for the *preference strength* to rank models**: given a triple $(x, y_{1}, y_{2})$ where all raters agree that $y_{1}$ is preferable over $y_{2}$, LL allows us to measure that a model A predicting $p_{A} = 0.9$ ($LL = -0.1$) is better (in that data point) than another model B predicting $p_{B} = 0.6$ ($LL = -0.5$). Accuracy would provide an equal score for both models since it only accounts for the binarized prediction. LL provides a more "fine-grained" measure.
 >
 >  Another crucial point is that **LL factors in the aleatoric uncertainty in the label-generating process**. For instance, in a scenario where only 70% of the raters agree that $y_{1}$ is preferable, LL better ranks models whose predictions are closer to p = 0.7, respecting the ground truth preference strength, which is not possible with accuracy.
 >  We also empirically illustrate in Figure 13 (Rebuttal PDF) in a simple problem setting that preference models with higher average LL lead to finetuned policies with higher win rates over the base policy (see **Q5**).

**Q2** What is the computational cost of BAL-PM? **(aVyo, tUWV)**
> **A2** We clarify this point since we argue that computational tractability is one of the main contributions of our method. First, some context: our work focuses on *(Bayesian) Active Learning*, which is naturally more computationally demanding than simply training predictive models. This is because **we require models that expresses epistemic uncertainty** to acquire informative labels for efficient training. This also **requires models to constantly update their uncertainties given the new data, via re-training**. The key is that **Active Learning reduces the number of labels required to train a better model, which considerably overcomes the additional computational cost**. The labeling process is considerably more expensive and laborious.
>
> As described in the Introduction, Preference Modeling in LLMs requires batch acquisition - it is impossible to request the label of a single point, re-train the model, and repeat this process. Still, tractable methods rely on these single-point acquisition objectives. Thus, **what BAL-PM does computationally is to replace $B - 1$ model re-trainings per acquired batch with computing entropy estimates** (considerably cheaper, as explained below). $B$ is the batch size, and $B = 320$ in our experiments.
>
> **BAL-PM does not require training or inference on LLMs during the active learning loops**. This considerably reduces the computational cost and allows us to scale up to 140b models in a single A100 GPU. To put it in perspective, fully fine-tuning a 7b model currently requires at least 4 A100s. LoRA methods also require new LLM inferences for every model update, while BAL-PM only requires once.
>
> The computation of BAL-PM has three pieces: offline processing (LLM inference and kNN computation), adapters update, and entropy estimation. LLM inference is done only once, prior to Active Learning. It is the bare minimum for LLM adoption. Furthermore, **we can compute the features used for the preference model and for entropy estimation in the same forward pass**: every prompt-answer input concatenates prompt/answer texts; thus, we can extract prompt features as the last layer embedding right after the last prompt token, and the prompt-answer features right after the answer's last token. Hence, there is no extra cost to extract features for entropy estimation.
> The cost of updating adapters is minimal, as they are MLPs with 2 hidden layers, reasonably cheap for LLM research. The entropy estimation only requires computing the di-gamma function (Equation 11) in the pool.
>
> Ultimately, our experiments show that BAL-PM can handle a challenging real-world preference dataset (used in the precursor of ChatGPT) with LLMs with up to 140b parameters, demonstrating its scalability (as highlighted by Reviewer x99G). We hope this clarifies its applicability in practical settings.

---

### Author Response · Authors · 2024-08-07
**Global Rebuttal References**

[1] Kossen et. al. Active Testing: Sample–Efficient Model Evaluation. ICML, 2021.

[2] Lakshminarayanan et. al. Simple and Scalable Predictive Uncertainty Estimation using Deep Ensembles. NeurIPS, 2017.

[3] Gal et. al. Dropout as a Bayesian Approximation: Representing Model Uncertainty in Deep Learning. ICML, 2016.

---

### Decision · Program_Chairs · 2024-09-25

**Decision:**

Accept (poster)

**Comment:**

This paper proposes to use active learning techniques to select examples for querying preference ratings for use in preference modeling algorithms. The main contention of the author is that paying attention only to epistemic uncertainty can result in collecting redundant batches of data and thus propose to also pay attention to the diversity of the data as characterized by "maximiz[ing] the entropy of the acquired prompt distribution in the feature space". Reviewers found the results to be strong/impressive/comprehensive, to be shown to scale, the paper to be clearly written. They also raised some concerns about the metric which doesn't actually show that the resulting data here led to more effective training of LLMs and the "link to the real-world problem of constructing a preference model in a sample-efficient fashion in the absence of a pool of labelled prompt-completion pairs". Following rebuttals, the mean score of the paper is a 6 and given the presence of champions and lack of a strong detractor I am recommending acceptance.